# STAT3 inhibition recovers regeneration of aged muscles by restoring autophagy in muscle stem cells

Giorgia Catarinella[1,*], Andrea Bracaglia[1,2,*], Emilia Skafida[1,3], Paola Procopio[1], Veronica Ruggieri[4,5], Cristina Parisi[4,5], Marco De Bardi[1], Giovanna Borsellino[1], Luca Madaro[4,5], Pier Lorenzo Puri[6], Alessandra Sacco[6], Lucia Latella[1,7]

Age-related reduction in muscle stem cell (MuSC) regenerative capacity is associated with cell-autonomous and non–cell-autonomous changes caused by alterations in systemic and skeletal muscle environments, ultimately leading to a decline in MuSC number and function. Previous studies demonstrated that STAT3 plays a key role in driving MuSC expansion and differentiation after injury-activated regeneration, by regulating autophagy in activated MuSCs. However, autophagy gradually declines in MuSCs during lifespan and contributes to the impairment of MuSC-mediated regeneration of aged muscles. Here, we show that STAT3 inhibition restores the autophagic process in aged MuSCs, thereby recovering MuSC ability to promote muscle regeneration in geriatric mice. We show that STAT3 inhibition could activate autophagy at the nuclear level, by promoting transcription of autophagy-related genes, and at the cytoplasmic level, by targeting STAT3/PKR phosphorylation of $eIF2\alpha$. These results point to STAT3 inhibition as a potential intervention to reverse the age-related autophagic block that impairs MuSC ability to regenerate aged muscles. They also reveal that STAT3 regulates MuSC function by both transcription-dependent and transcription-independent regulation of autophagy.

## Introduction

The regenerative potential of skeletal muscle relies on the presence of muscle stem cells (MuSCs) located underneath the basal lamina in a quiescent state (Evano & Tajbakhsh, 2018). In response to stress or injury, MuSCs exit the quiescence state, undergo proliferation, self-renew, and eventually differentiate to repair the tissue (Collins et al, 2005). During aging, muscle regenerative functions decline, pointing to MuSCs as a key dysfunctional effector of age-associated disruption of muscle homeostasis (Blau et al, 2015). This failure results from cell-autonomous alterations in MuSC properties and from extrinsic/systemic alterations in an aged niche (Chakkalakal et al, 2012; Dumont et al, 2015). Among the cell-intrinsic signals that contribute to age-dependent impairment of MuSC ability to regenerate injured muscles, a persistent increase in inflammatory cytokines appears to play a major role. For instance, increased circulating levels of IL-6 are typically detected in aged organisms (Muñoz-Cánoves et al, 2013) and increased IL-6 expression has been reported in geriatric MuSCs (Sousa-Victor et al, 2014) and in other muscle-resident cells in conditions leading to muscle atrophy (Haddad et al, 2005; Zhang et al, 2013; Madaro et al, 2018). We and others have previously demonstrated a key role of IL-6/JAK2/STAT3 signaling in regulating MuSC expansion during muscle regeneration (Price et al, 2014; Tierney et al, 2014). In particular, we showed that the IL-6/STAT3 pathway controls MuSC expansion and differentiation during muscle tissue repair in young, aged, and dystrophic conditions (Tierney et al, 2014). Likewise, STAT3 inhibition promotes symmetric expansion of MuSCs in vitro and increases their muscle engraftment upon transplantation in vivo (Price et al, 2014). These results provide the rationale for pharmacological inhibition of STAT3 as a potential approach to counter impaired regeneration and counter muscle atrophy during pathological conditions and aging.

Protein quality control and metabolic pathways represent important processes to guarantee adult cell stem quiescence and homeostasis (García-Prat et al, 2017). Notably, all these processes decline with aging and can be manipulated to reestablish the functionality of the stem cell compartment and obtain an anti-aging effect (Cuervo et al, 2005). Macroautophagy is an intracellular degradation system by which dysfunctional organelles and proteins are engulfed into lysosomes. This process not only eliminates material but also functions as an intracellular quality check and

[1]IRCCS Fondazione Santa Lucia, Rome, Italy [2]PhD Program in Cellular and Molecular Biology, Department of Biology, University of Rome "Tor Vergata", Rome, Italy [3]School of Medicine and Surgery, University of Milano-Bicocca, Monza, Italy [4]Department of Anatomy, Histology, Forensic Medicine and Orthopedics, University of Roma "La Sapienza", Rome, Italy [5]Laboratory Affiliated to Istituto Pasteur Italia-Fondazione Cenci Bolognetti, Rome, Italy [6]Development, Aging and Regeneration Program, Sanford Burnham Prebys Medical Discovery Institute, La Jolla, CA, USA [7]Institute of Translational Pharmacology, National Research Council of Italy, Rome, Italy

Correspondence: lucia.latella@ift.cnr.it
*Giorgia Catarinella and Andrea Bracaglia contributed equally to this work

as a dynamic recycling system to produce energy for cellular renovation and homeostasis (Mizushima & Komatsu, 2011; Singh & Cuervo, 2011). Recent studies unveiled the pivotal role of the autophagic process in efficient activity of MuSCs throughout the different phases of the repair process. For instance, autophagy provides the nutrients necessary to meet the bioenergetic demand of MuSCs to face the transition from the quiescence to the activation phase (Tang & Rando, 2014). Indeed, inhibition of autophagy is sufficient to delay MuSC activation (Tang & Rando, 2014). During MuSC activation, metabolic changes occur alongside an increased need for protein biosynthesis and macromolecules (Mizushima et al, 2004). Accordingly, we showed that the autophagic process is activated in MuSCs during muscle regeneration and significantly declines during aging (Fiacco et al, 2016). Moreover, autophagy serves as a cytoprotective and quality control mechanism to balance protein and organelle turnover during homeostasis proving essential for MuSC stemness and functionality (García-Prat et al, 2016). Age-associated failure of autophagy leads to the accumulation of misfolded proteins, toxic debris, and damaged organelles in MuSCs. This "proteotoxic" stress produces elevated levels of reactive oxygen species, DNA damage, and the derepression of *P16INK4a* resulting in MuSCs entering a senescence state (García-Prat et al, 2016). Different approaches aimed at reactivating the autophagic flux extend lifespan and regulate stem cell function, highlighting the pivotal role autophagy plays in ensuring efficient muscle repair (Cerletti et al, 2012; Yang et al, 2016).

Growing evidence supports the notion that STAT3 plays a significant role in the autophagic regulation (Kroemer et al, 2010). This involvement can result in either the activation or inhibition of autophagy, depending on the cell type, cellular context, and the intracellular localization (You et al, 2015). STAT3 has been implicated in both the initial formation of the autophagosome and its final maturation within the catabolic process of skeletal muscle (You et al, 2015). Based on its localization, STAT3 can modulate autophagy in both a transcription-dependent and a transcription-independent manner. On the one hand, nuclear STAT3 fine-tunes autophagy via the transcriptional regulation of several autophagy-related genes such as the BCL2 family members (*Bcl2, Bcl2L1,* and *Mcl1*), *Bnip3* (Pratt & Annabi, 2014), *Becn1* (Miao et al, 2014), and *Pik3c3* (*Vsp34*) (Yamada et al, 2012). On the other hand, cytoplasmic STAT3 constitutively represses autophagy by sequestering protein kinase R (PKR) and preventing the phosphorylation of eukaryotic initiation factor 2$\alpha$ (eIF2$\alpha$) (Shen et al, 2012). eIF2$\alpha$ phosphorylation represses global translation but allows the preferential translation of activating transcription factor 4 (ATF4), a regulator that controls the transcription of key genes essential for adaptive functions, eventually activating the autophagic process (Shen et al, 2012). Likewise, cytoplasmic STAT3 displays an anti-autophagic function by interacting with other autophagy-related signaling molecules such as FOXO1 and FOXO3 (Mammucari et al, 2007). In addition, STAT3 translocation into mitochondria suppresses the reactive oxygen species–induced autophagy induced by oxidative stress preserving mitochondria from being degraded by mitophagy (Szczepanek et al, 2011).

Here, we show that STAT3 inhibition directly activates the autophagic flux in MuSCs, eventually promoting regeneration both in young and in aged muscles. This is of particular relevance,

especially in the context of aging, because STAT3 signaling is elevated in aged MuSCs and antagonizes their symmetric expansion (Price et al, 2014). At the same time, the decline in the autophagic process throughout lifespan compromises the MuSC functionality (García-Prat et al, 2016). We therefore tested whether STAT3 inhibition could restore MuSC-mediated regeneration of aged muscles by reversing STAT3-mediated inhibition of autophagy.

# Results

## STAT3 inhibition leads to increased autophagy in young and old regenerating muscles

To assess the role of STAT3 in regulating the autophagic process during muscle regeneration, we treated cardiotoxin (CTX)-injured WT mice with a specific STAT3 inhibitor (STAT3i). The STAT3 inhibitor (#573096; Sigma-Aldrich) used is a cell-permeable peptide, an analog of the STAT3-SH2 domain–binding phosphopeptide, that acts as a selective blocker of the STAT3 activity. We administered the STAT3i according to the concentration and conditions described in our previous work (Tierney et al, 2014). Thus, the STAT3i was injected intramuscularly (i.m.) in tibialis anterior (TA) muscles of 2-mo-old (young) or 15-mo-old (old) WT mice, and muscles were harvested 3 d post-injury (d.p.i.) (Fig 1A). In these experimental conditions, the transient inhibition of STAT3 promotes the expansion of MuSCs (Tierney et al, 2014). Before examining the impact of treatment on autophagy, we first measured the STAT3 levels in young and old skeletal muscles and tested the efficacy of STAT3 inhibition. We detected increased levels of STAT3 in aged mice compared with their young counterparts (Fig 1B). In both cases, we verified that the inhibitor was efficient at the concentration and conditions used, as shown by the reduced levels of STAT3 tyrosine 705 phosphorylation upon STAT3i administration (P-STAT3$^{Y705}$) (Fig 1B). Next, to measure the autophagic flux, we administered chloroquine (CLQ) 4 h (hrs) before euthanasia. In particular, CLQ hampers the binding of autophagosomes to lysosomes by altering the acidic environment of lysosomes, thus blocking the autophagic flux (Perry et al, 2009). As a consequence, upon CLQ treatment, lipidated LC3 increases exclusively when the autophagic process is active. By measuring the relative change in the LC3II/GAPDH ratio between CLQ-treated and not treated samples ($\Delta$LC3II/GAPDH), we showed that the autophagic flux increases upon STAT3i treatment in young mice (Fig 1C). As an additional marker to study the autophagic flux, we monitored the ubiquitin-binding scaffold protein p62/SQSTM1 that accumulates when autophagy is inhibited and decreases when autophagy is induced. We detected lower levels of p62/SQSTM1 upon STAT3i administration confirming the activation of the autophagic process (Fig 1D). At the transcriptional level, we detected an up-regulation of *p62* after STAT3i treatment (Fig 1E). A transcriptional increase in *p62* has been previously reported to be an adaptive stress response (Sahani et al, 2014), and *p62* transcriptional activation has been shown to be dependent on the eIF2$\alpha$/ATF4 pathway (B'chir et al, 2013).

Following the same experimental pipeline (Fig 1A), we in addition analyzed the impact of STAT3 inhibition on 15-mo-old (old) mice

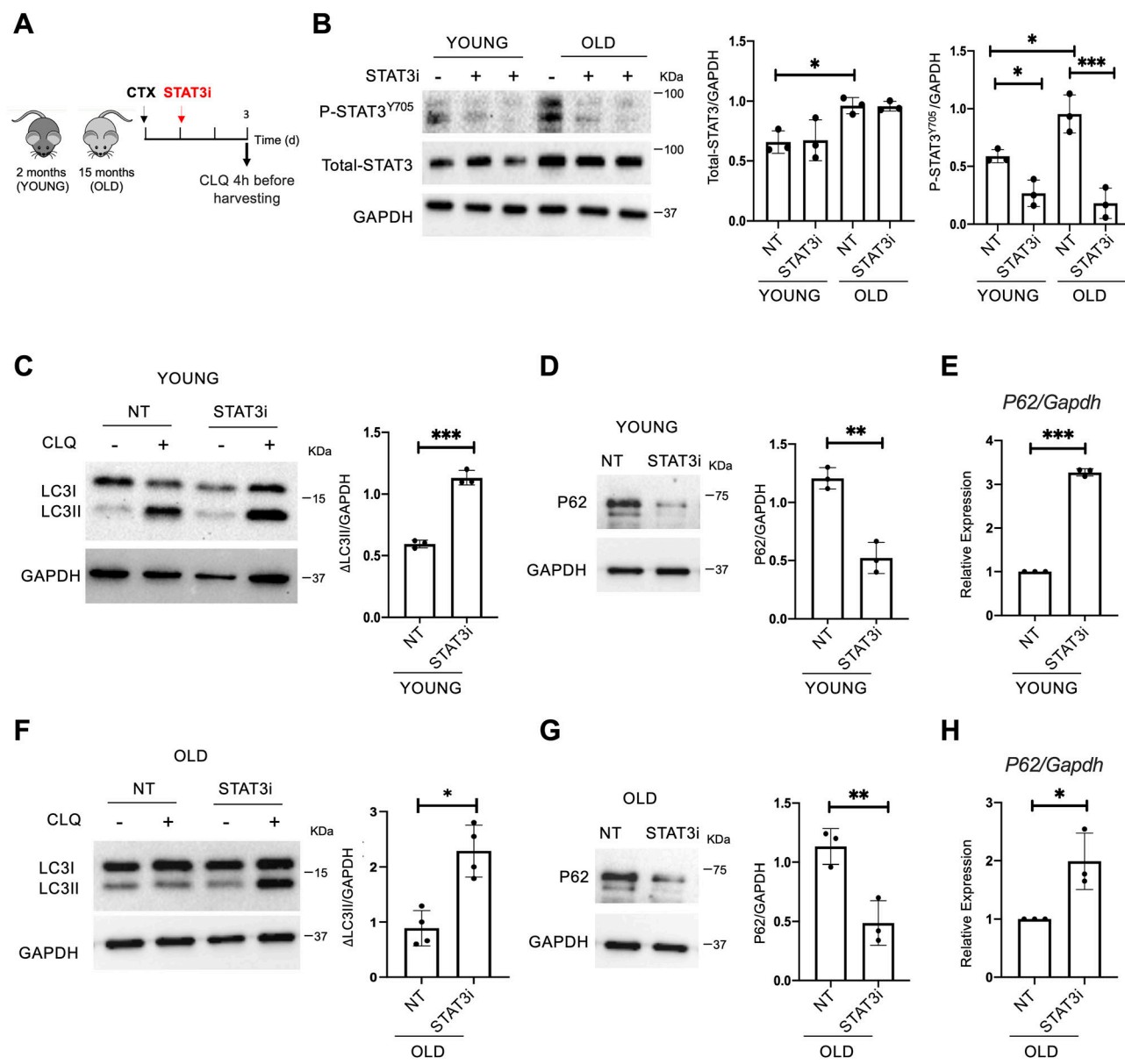

**Figure 1. STAT3 inhibition activates autophagy in young and old regenerating muscles.**
**(A)** Schematic representation of skeletal muscle injury and STAT3i treatment in 2 (young)- or 15 (old)-month-old C57BL/6J WT mice. To assess the autophagic flux, CLQ was administered 4 h before harvesting. **(B)** Phospho-STAT3$^{Y705}$ (P-STAT3$^{Y705}$), total STAT3, and GAPDH protein expression in young or old uninjured muscles. Plots represent the total STAT3/GAPDH ratio and the P-STAT3$^{Y705}$/GAPDH ratio (n = 3; values represent the mean ± s.d., *$P$ < 0.05 and ***$P$ < 0.001 by one-way ANOVA). **(C)** LC3 protein expression in young injured WT mice analyzed at 3 d.p.i. The plot represents the relative change in the LC3II/GAPDH ratio between CLQ-treated and not treated samples (ΔLC3II/GAPDH) (n = 3; values represent the mean ± s.d., ***$P$ < 0.001 by a $t$ test). **(D)** P62 protein expression in young mice not treated (NT) or treated with the STAT3i. The plot represents the p62/GAPDH ratio (n = 3; values represent the mean ± s.d., **$P$ < 0.01 by a $t$ test). **(E)** qRT–PCR for $p62$ normalized for $Gapdh$ in young mice not treated (NT) or treated with the STAT3i (n = 3; values represent the mean ± s.d., ***$P$ < 0.001 by a $t$ test). **(F)** LC3 protein expression in old injured WT mice analyzed at 3 d.p.i. The plot represents the ΔLC3II/GAPDH ratio (n = 4; values represent the mean ± s.d., *$P$ < 0.05 by a $t$ test). **(G)** P62 protein expression in old mice not treated (NT) or treated with the STAT3i. The plot represents the p62/GAPDH ratio (n = 3; values represent the mean ± s.d., **$P$ < 0.01 by a $t$ test). **(H)** qRT–PCR for $p62$ normalized for $Gapdh$ in old mice not treated (NT) or treated with the STAT3i (n = 3; values represent the mean ± s.d., *$P$ < 0.05 by a $t$ test).
Source data are available for this figure.

that feature low levels of autophagy (Fiacco et al, 2016), loss of proteostasis, increased mitochondrial dysfunction, and oxidative stress (García-Prat et al, 2016). Again, we evaluated the ability of the STAT3i to regulate autophagy by measuring the ΔLC3II/GAPDH (Fig 1F), the p62/SQSTM1 protein (Fig 1G), and mRNA levels (Fig 1H) in

control versus treated mice. Interestingly, we observed that the inhibition of STAT3 could restore the autophagic flux in old mice. Altogether, these data show that the administration of the STAT3i activates autophagy in regenerating muscle tissues in both young and old mice. In both contexts, STAT3i-mediated autophagy was

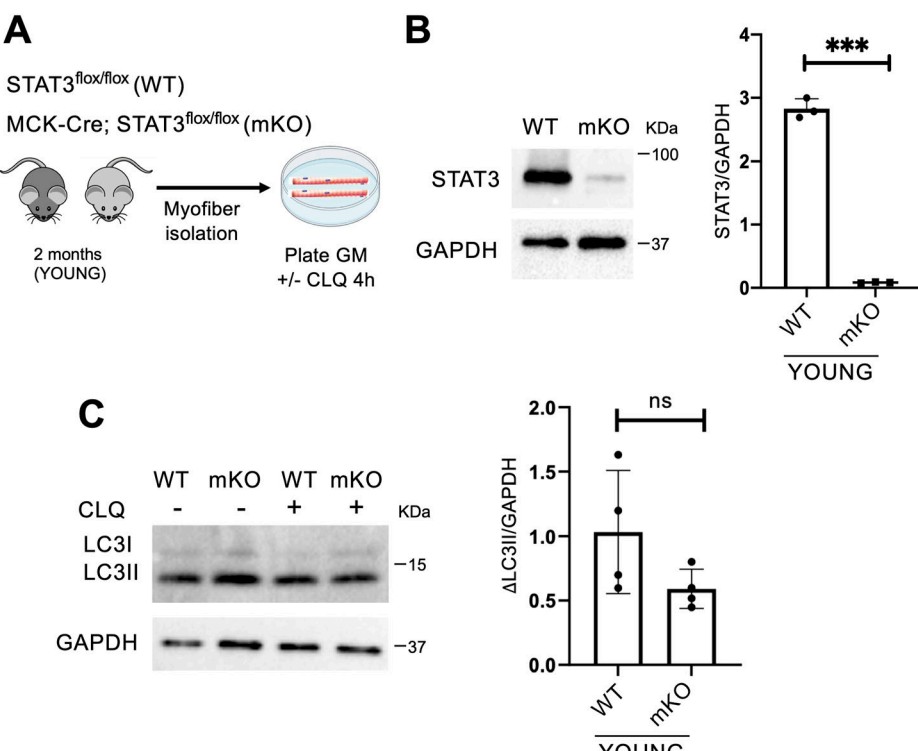

**Figure 2. STAT3 deletion in myofibers has no impact on the autophagic process.**
**(A)** Schematic representation of the ex vivo experiment in myofibers isolated from young STAT3f/f (WT) or MCK-Cre;STAT3f/f mice (mKO), and cultured in growth medium (GM). To evaluate the autophagic flux, CLQ was administered 4 h before harvesting. **(B)** STAT3 protein expression in isolated myofibers from WT and mKO mice. The plot represents the STAT3/GAPDH ratio (n = 3; values represent the mean ± s.d., ***P < 0.001 by a *t* test). **(C)** LC3 protein expression in isolated myofibers from WT and mKO mice. The plot represents the relative change in the LC3II/GAPDH ratio between CLQ-treated and not treated samples (ΔLC3II/GAPDH) in WT and mKO mice (n = 4; values represent the mean ± s.d., not significant [ns] by a *t* test).

associated with an increased number of PAX7-positive MuSCs (Fig S1A and B) and more efficient muscle regeneration as assessed by cross-sectional area measurements, embryonal MyHC-positive myofibers, and hematoxylin and eosin staining (Fig S2A–D).

### Genetic or pharmacological inactivation of STAT3 affects the autophagic process in MuSCs but not in myofibers

The interpretation of the outcome of STAT3 inhibition administered by i.m. injection is complicated by the fact that virtually all cell types in the regenerative milieu of the whole muscles could be targeted by the STAT3i. To overcome this issue, we first investigated whether the effect of STAT3 inhibition on the autophagic process was accounted by the most represented cell type within skeletal muscles, that is, the myofibers. To this purpose, we employed the MCK-Cre;STAT3f/f mice to genetically delete STAT3 in myofibers (mKO) compared with the STAT3f/f mice (WT). We isolated myofibers from the extensor digitorum longus (EDL) muscles, and we cultured them ex vivo in growth medium (GM) for 4 h with or without CLQ to determine the autophagic flux (Fig 2A). We first verified the ablation of STAT3 in myofibers isolated from STAT3f/f or MCK-Cre;STAT3f/f mice by Western blot for STAT3 (Fig 2B). Next, we assessed the autophagic flux by measuring the relative change in the LC3II/GAPDH ratio between CLQ-treated and not treated samples (ΔLC3II/GAPDH). No significant changes in ΔLC3II/GAPDH were observed in myofibers either from STAT3f/f or from MCK-Cre;STAT3f/f (Fig 2C). This evidence supports the

conclusion that STAT3 can regulate autophagy in skeletal muscles either from myofibers (e.g., by modulating paracrine signals) or from other muscle-resident cell types.

Given that previous works have shown that STAT3 regulates MuSC biology at multiple levels (Price et al, 2014; Tierney et al, 2014; Zhu et al, 2016; Sala et al, 2019), we evaluated the effect of STAT3 depletion in MuSCs, using Pax7-CreER;STAT3f/f mice. 2-mo-old Pax7-CreER;STAT3f/f mice were treated with TMX for 5 d and subsequently injured, and MuSCs were freshly isolated 3 d.p.i. (Fig 3A). We confirmed the efficiency of the genetic deletion by analyzing *Stat3* transcriptional levels and its downstream target *MyoD* that we found both down-regulated (Fig 3B). To eliminate the potential impact of other muscle-resident cell types, we evaluated the effect of STAT3 deletion ex vivo using single myofibers isolated from Pax7-CreER;STAT3f/f mice (Fig 3C). TMX treatment increased autophagy in MuSCs (as detected by PAX7[+]/LAMP1[+] cells) only at an early time point (i.e., 48 h) (Fig 3D), leading to their expansion at a later time point (i.e., 60 h) (Fig 3E). Because LAMP1 is a marker of the lysosomal content (Eskelinen, 2006), we performed an additional immunostaining in myofibers with the canonical autophagic marker LC3. We detected an increase in LC3 in PAX7[+] cells upon STAT3 inhibition (Fig 3F and G) (Bisicchia et al, 2022). A similar effect was observed in myofibers isolated from WT mice ex vivo treated with the STAT3i (Fig S3A–D). These data support the notion that STAT3 inhibition in MuSCs, either by genetic or by pharmacological deletion, activates the autophagic process, followed by MuSC proliferation and expansion. Accordingly, an increased number of MuSC nuclei clustered within isolated single myofibers were observed upon either

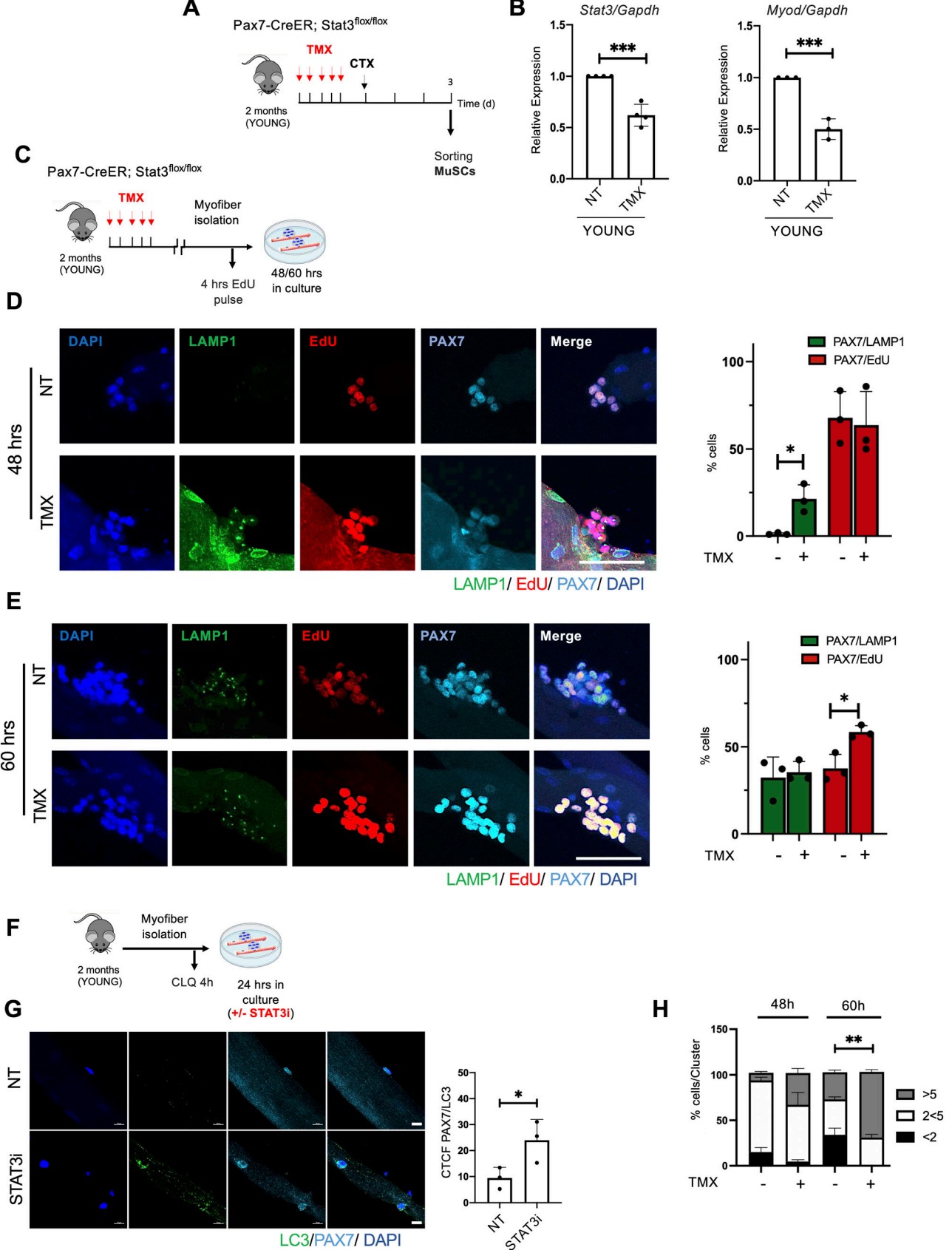

TMX treatment of Pax7-CreER;STAT3f/f mice (Fig 3H) or STAT3i treatment in myofibers isolated from WT mice (Fig S3D).

### STAT3 inhibition recovers the autophagic process in old MuSCs

The finding that the activation of autophagy precedes induction of MuSC proliferation suggests a potential temporal relationship between these two events. This prompted an interest to evaluate whether STAT3 inhibition could be exploited to unlock autophagy in MuSCs from old mice (García-Prat et al, 2016). To this aim, 2-mo-old (young) and 15-mo-old (old) WT mice were injured and treated with the STAT3i or vehicle as a control (NT). MuSCs were isolated 3 d.p.i., and CLQ was administered on digested muscles 4 h before FACS isolation (Fig 4A) (Campanario et al, 2021). MuSCs were assayed immediately after isolation. We first validated the efficiency of the STAT3i by monitoring the gene expression of *Socs3* and *MyoD*, two STAT3 target genes (Fig 4B). Upon STAT3i treatment, we detected the activation of the autophagic flux in young (Fig 4C—left panels) and old (Fig 4C—right panels) MuSCs as measured by the increase in ΔLC3II/GAPDH.

We further analyzed the autophagic process by the immuno-fluorescence staining for LAMP1 to monitor the formation of autolysosomes in MuSCs not treated and treated with the STAT3i (Fig 4D). We implemented these data with a co-staining for LC3 and LAMP1 demonstrating co-localization of the two proteins (Fig 4E), confirming the boost of autophagy in young and the restoration of the autophagic process in old MuSCs upon STAT3 inhibition.

Using the Pax7-CreER;STAT3f/f as a mouse model for STAT3 gene deletion in MuSCs (Fig 4F), we found that STAT3 removal is sufficient to activate the autophagic flux in MuSCs, irrespective of whether they were isolated from young or old mice (Fig 4G). These data indicate that STAT3 negatively regulates autophagy in MuSCs throughout the lifespan, with the inhibition of autophagy conferring to MuSCs an age-associated impairment in regenerative response to injury, a condition that can be overcome by STAT3 inhibition.

### Cytoplasmic translocation of STAT3 after STAT3i treatment is coupled with autophagy activation in young and old MuSCs

Increasing evidence shows that STAT3 is an autophagy regulator not only at the nuclear level, by regulating gene expression, but also by virtue of its cytoplasmic localization (You et al, 2015). To unveil the mechanism by which the STAT3i promotes/resumes the autophagic process in young and old MuSCs toward an efficient regenerative response, we monitored STAT3 intracellular localization.

To this purpose, we exposed young and old WT mice to STAT3i treatment followed by MuSC isolation at 3 d.p.i. (Fig 5A). Right after FACS isolation, young and old MuSCs were co-stained for STAT3 and LAMP1. We show that upon STAT3 inhibition, STAT3 re-localizes toward the cytoplasm. Of note, in both young and old MuSCs, cytoplasmic STAT3 accumulation was invariably associated with activation of autophagy (Fig 5B and C), suggesting that STAT3 might be directly involved in regulating the autophagic process. Noteworthy, when we estimated the relative content in nuclear versus total STAT3 intensity, we detected a significant increase in cytoplasmic STAT3 in old MuSCs (Fig 5D), supporting the hypothesis that in aged MuSCs, cytoplasmic STAT3 constitutively inhibits autophagy by sequestering PKR, as described by Shen and colleagues (Shen et al, 2012). It is worth emphasizing that this measurement did not take into account the total amount of STAT3—that is higher in aged MuSCs—but only the relative ratio of nuclear to total STAT3. Upon STAT3i administration, STAT3 was no longer phosphorylated in Tyr705, which prevents its nuclear translocation. This resulted in the reduction of nuclear versus total STAT3 intensity upon the STAT3i in both young and old MuSCs (Fig 5D). At the same time, the STAT3i disrupts the STAT3/PKR binding, allowing PKR to phosphorylate eIF2α and activating autophagy.

To investigate whether cytoplasmic STAT3 regulates the autophagic process in muscle progenitors, we employed the C2C12 mouse myoblast cell line. In particular, we evaluated the ability of the STAT3i to dissociate the STAT3/PKR complex allowing the PKR-mediated phosphorylation of eIF2α. The rationale behind the use of C2C12 lies in the need for adequate amounts of cytoplasmic extracts that are otherwise difficult to obtain with MuSCs, as they have a reduced cytoplasmic/nuclear ratio. C2C12 myoblasts were treated in growth media with the STAT3i for 24 h at the concentration of 50 µM. The significant reduction of P-STAT3$^{Y705}$ and MyoD levels was estimated as a suitable readout to control the effectiveness of the STAT3i treatment (Fig S4A), as well as the down-regulation of the STAT3 target genes *Socs3* and *MyoD* (Fig 6A). Interestingly, we found increased levels of pro-autophagic genes (e.g., *LC3*, *Atg9A*, *Ulk1*, *Atg7*, *Bnip3*, *Atg4*, and *Becn1*), whereas the autophagy inhibitor *Bcl2* was reduced by the STAT3i (Fig 6B). This suggests that STAT3 inhibition promotes the activation of pro-autophagic genes to support the activation of the autophagic process. A common transcriptional signature consisting of the activation of autophagy-related genes (e.g., *LC3*, *Atg9A*, *Ulk1*, *Bnip3*, and *Becn1*) was detected in C2C12 and MuSCs upon STAT3 inhibition (Fig S5C).

To measure the autophagic flux, cells were treated with CLQ at the dosage of 30 µM 4 h before cell harvesting. We observed an increase, compared with a basal level, in the autophagic flux in

---

**Figure 3. Deletion of STAT3 in muscle stem cells (MuSCs) induces the autophagic process, which precedes the proliferative expansion phase.**
**(A)** Schematic representation of TMX treatment and skeletal muscle injury in 2-mo-old (young) Pax7-CreER;STAT3f/f mice. MuSCs were isolated 3 d.p.i. **(B)** qRT–PCR for *Stat3* and *MyoD* in MuSCs normalized for *Gapdh* (n = 4; values represent the mean ± s.d., ***P < 0.001 for *Stat3* by a *t* test; n = 3; values represent the mean ± s.d., ***P < 0.001 for *MyoD* by a *t* test). **(C)** Schematic representation of the TMX treatment of young Pax7-CreER;STAT3f/f mice and myofiber isolation. Isolated myofibers were cultured ex vivo for 48 or 60 h and exposed to 4-h EdU pulse before harvesting. **(D, E)** Representative images of LAMP1 (green), EdU (red), PAX7 (cyan), and DAPI (blue) immunostaining in myofibers harvested at 48 h (D) and 60 h (E) of culture. Scale bar = 50 µm. Plots represent the percentage of PAX7$^+$/LAMP1$^+$ (green bars) and PAX7$^+$/EdU$^+$-positive (red bars) cells (n = 3; 15 myofibers per biological sample were counted; values represent the mean ± s.d., *P < 0.05 by a *t* test). **(F)** Schematic representation of myofiber ex vivo treatment with the STAT3i for 24 h. To assess the autophagic flux, CLQ was administered 4 h before harvesting. **(G)** Representative images of myofibers immunostained for LC3 (green), PAX7 (cyan), and DAPI (blue). Scale bar = 10 µm. The plot represents the corrected total cell fluorescence of LC3 over the area of the cell (n = 3; 15 myofibers per biological sample were counted; values represent the mean ± s.d., *P < 0.05 by a *t* test). **(H)** Plot represents the percentage of cells for each cluster of satellite cells on the myofiber (%cells/Cluster) at 48 and 60 h post-isolation (n = 3; values represent the mean ± s.d., **P < 0.01 by a *t* test).

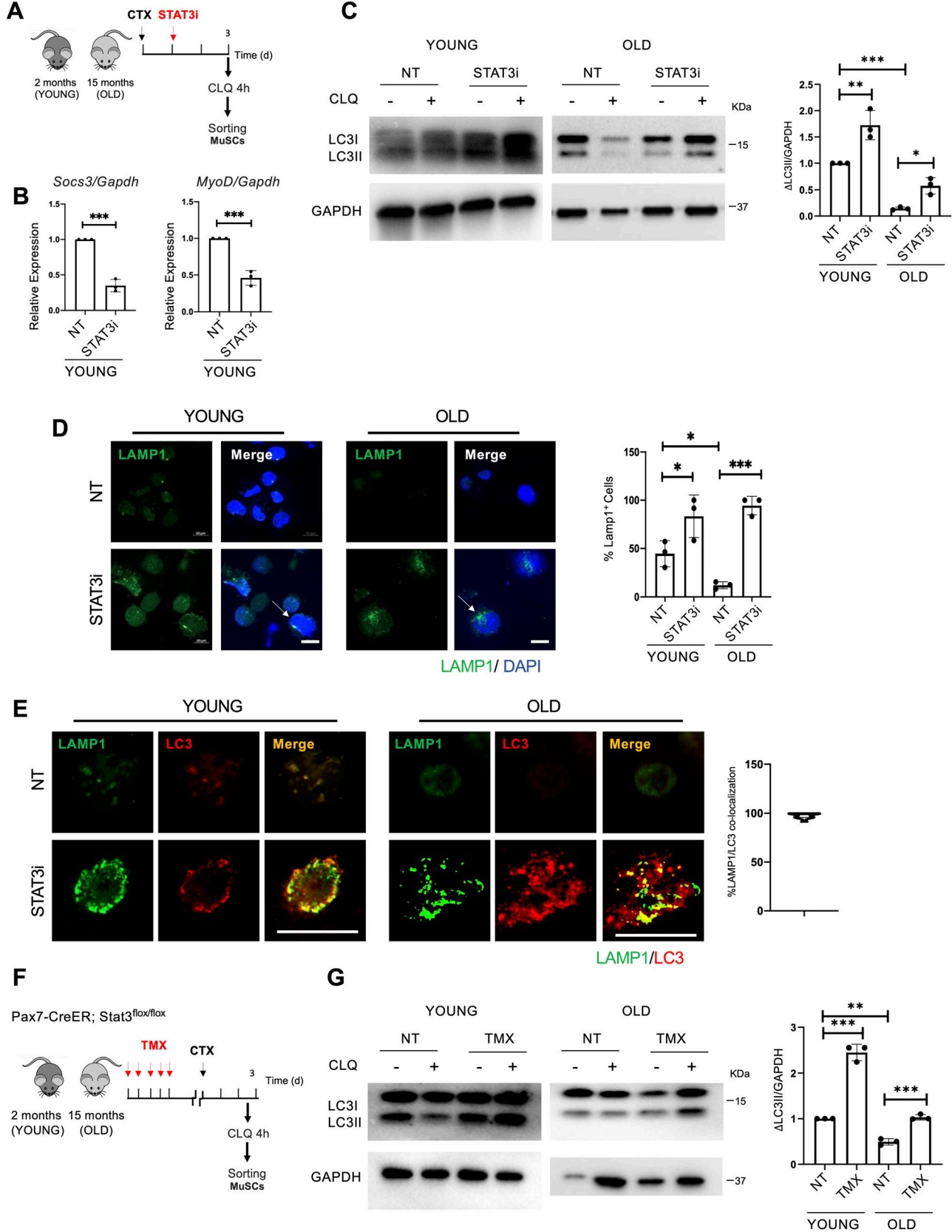

C2C12 exposed to STAT3i treatment, as revealed by the increase in the accumulation of LC3 puncta ($\Delta$LC3 puncta/cell) (Fig S4B) and in the amount of lipidated LC3 ($\Delta$LC3II/GAPDH) in the presence of the lysosome inhibitor (Fig S4C). We also detected an increase in phosphorylated eIF2$\alpha$ upon STAT3 inhibition (Fig 6C) that triggered us to further investigate the impact of the treatment on the STAT3/PKR complex stability. Therefore, we performed an immunoprecipitation (IP) assay for STAT3 in the cytoplasmic extracts of C2C12 cells treated with the STAT3i. Interestingly, IP analysis showed that in the cytoplasm, STAT3 binds PKR in untreated cells, whereas this interaction is significantly decreased upon STAT3i treatment (Fig 6D), concurrently with autophagy induction (Fig S4B and C). The disruption of the STAT3/PKR complex upon the STAT3i was further confirmed by a proximity ligation assay (PLA) that revealed a nearly absent signal when C2C12 cells were treated with the STAT3i (Fig 6E). Likewise, disruption of the STAT3/PKR complex upon the STAT3i was also detected in MuSCs (Fig S5A and B).

# Discussion

Collectively, our results demonstrate that STAT3 regulates autophagy during MuSC activation. Constitutive activation of STAT3 is associated with reduced autophagy in aged MuSCs. Inhibition of STAT3 could resume the autophagic flux to support the bioenergetic demand of activated MuSCs, uncovering a previously unappreciated link between STAT3 and reduced autophagy in MuSCs during aging. Interestingly, our data indicate that the recovery of the autophagic flux induced by STAT3 inhibition enables aged MuSCs to resume proliferation toward countering age-related reduction of muscle. Previous studies have reported on the beneficial effects of STAT3 inhibition in reversing the age-related decline in MuSC ability to promote regeneration (Price et al, 2014; Tierney et al, 2014; Zhu et al, 2016); likewise, impaired autophagy has been associated with loss of MuSC regenerative properties in geriatric mice (García-Prat et al, 2016). Here, we have identified STAT3 as a nodal factor that coordinates these events, whereby STAT3-mediated inhibition of autophagy negatively regulates MuSC ability to proliferate in aged muscles. Interestingly, restoration of the autophagy flux and proliferative ability of aged MuSCs by STAT3 inhibition appears to relate to a previously unrecognized cytoplasmic function of STAT3 in MuSCs.

During aging, skeletal muscle function and repair capabilities progressively decline, because of significant changes in tissue composition, which include increased inflammation and fibrosis and impaired tissue innervation, in association with alterations in proteostasis and tissue metabolism. Previous works showed that STAT3 signaling pathway activity is elevated during aging, along with the age-associated increase in its upstream regulator IL-6, and that its transient inhibition promotes MuSC expansion and tissue repair (Price et al, 2014; Tierney et al, 2014). The secreted factor Fam3a—a direct STAT3 downstream effector—regulates mitochondrial respiration in MuSCs (Sala et al, 2019). Autophagy is a major contributor to cellular metabolism, and several studies have demonstrated the pivotal role of autophagy in maintaining stem cell functionality (García-Prat et al, 2016), as a crucial quality control mechanism to guarantee cellular homeostasis (Guan et al, 2013). Indeed, quiescent stem cells attenuate the proteotoxicity by increasing the autophagic process to prevent the accumulation of toxic waste such as damaged organelles and proteins, becoming a "clean-up" process that prevents their premature entry into senescence and thereby their functional exhaustion (Hubbard et al, 2010; Tang & Rando, 2014; Fiacco et al, 2016). STAT3 can either activate or inhibit autophagy, depending on the cell type and the intracellular localization (You et al, 2015). Nuclear STAT3 predominantly represses autophagy by the transcriptional activation of *Bcl2* and *Mcl1* expression, which leads to autophagy inhibition, or by downregulation of *Becn1* and *Pik3C3*, which are associated with autophagy induction. Indeed, IL-6 treatment has been found to inhibit starvation-induced autophagy through the STAT3-mediated regulation of the Bcl2/Becn1 signaling pathway (Qin et al, 2015). Furthermore, although STAT3 has been classically characterized as a transcription factor, recent reports show that STAT3 can exert transcription-independent effects on the regulation of the autophagic process. It is of special interest that STAT3 regulates the subcellular localization of FoxO1 and FoxO3 controlling the duration of T-cell proliferation (Oh et al, 2012). In this regard, different data underlined the role of FoxO3 in controlling autophagy in skeletal muscle and in adult neural stem cells (Mammucari et al, 2007; Sanchez et al, 2012; Audesse et al, 2019). In particular, it has been found that FoxO3 controls the transcription of autophagy-related genes, *Lc3* and *Bnip3*. Furthermore, a novel mechanism through which cytoplasmic STAT3 inhibits the autophagic flux has been described in other cell types (Shen et al, 2012). In particular, latent cytoplasmic STAT3 binds to PKR kinase, inhibiting its activity and preventing eIF2$\alpha$ phosphorylation, eventually reducing autophagy. Hence, by inhibiting STAT3, the PKR kinase is available to phosphorylate eIF2$\alpha$ leading to an increase in the autophagic flux.

**Figure 4. STAT3 ablation resumes the autophagic process in old muscle stem cells (MuSCs).**
**(A)** Schematic representation of skeletal muscle injury and STAT3i treatment in 2 (young)- and 15 (old) month-old WT mice. MuSCs were isolated 3 d.p.i. To assess the autophagic flux, CLQ was administered 4 h before MuSC isolation. **(B)** qRT–PCR for *Socs3* and *MyoD* normalized for *Gapdh* (n = 3; values represent the mean ± s.d., ***$P < 0.001$ by a *t* test). **(C)** LC3 protein expression in MuSCs isolated from young or old mice, not treated (NT) or treated with the STAT3i. The plot represents the relative change in the LC3II/GAPDH ratio between CLQ-treated and not treated samples ($\Delta$LC3II/GAPDH) (n = 3; values represent the mean ± s.d., *$P < 0.05$, **$P < 0.01$, ***$P < 0.001$ by one-way ANOVA). **(D)** Representative images of LAMP1 (green) and DAPI (blue) immunostaining in MuSCs from young or old mice not treated (NT) or treated with the STAT3i. The arrows represent the accumulation of autolysosomes. Scale bar = 10 $\mu$m. The plot represents the percentage of LAMP1+ cells (n = 3; 30 MuSCs per biological sample were counted; values represent the mean ± s.d., *$P < 0.05$ and ***$P < 0.001$ by one-way ANOVA). **(E)** Representative images of LC3 (red) and LAMP1 (green) immunostaining in MuSCs isolated from young or old mice, not treated (NT) or treated with the STAT3i. The plot represents the percentage of LAMP1 and LC3 co-localization. Scale bar = 50 $\mu$m. The plot represents the percentage of LAMP1 and LC3 co-localization (n = 3; 20 MuSCs per biological sample were counted). **(F)** Schematic representation of TMX treatment and skeletal muscle injury in 2 (young)- or 15 (old)-month-old Pax7-CreER;STAT3f/f mice. MuSCs were isolated 3 d.p.i. To assess the autophagic flux, CLQ was administered 4 h before MuSC isolation. **(G)** LC3 protein expression in MuSCs isolated from young or old Pax7-CreER;STAT3f/f mice, not treated (NT) or treated with TMX. The plot represents the $\Delta$LC3II/GAPDH ratio (n = 3; values represent the mean ± s.d., **$P < 0.01$ and ***$P < 0.001$ by one-way ANOVA).

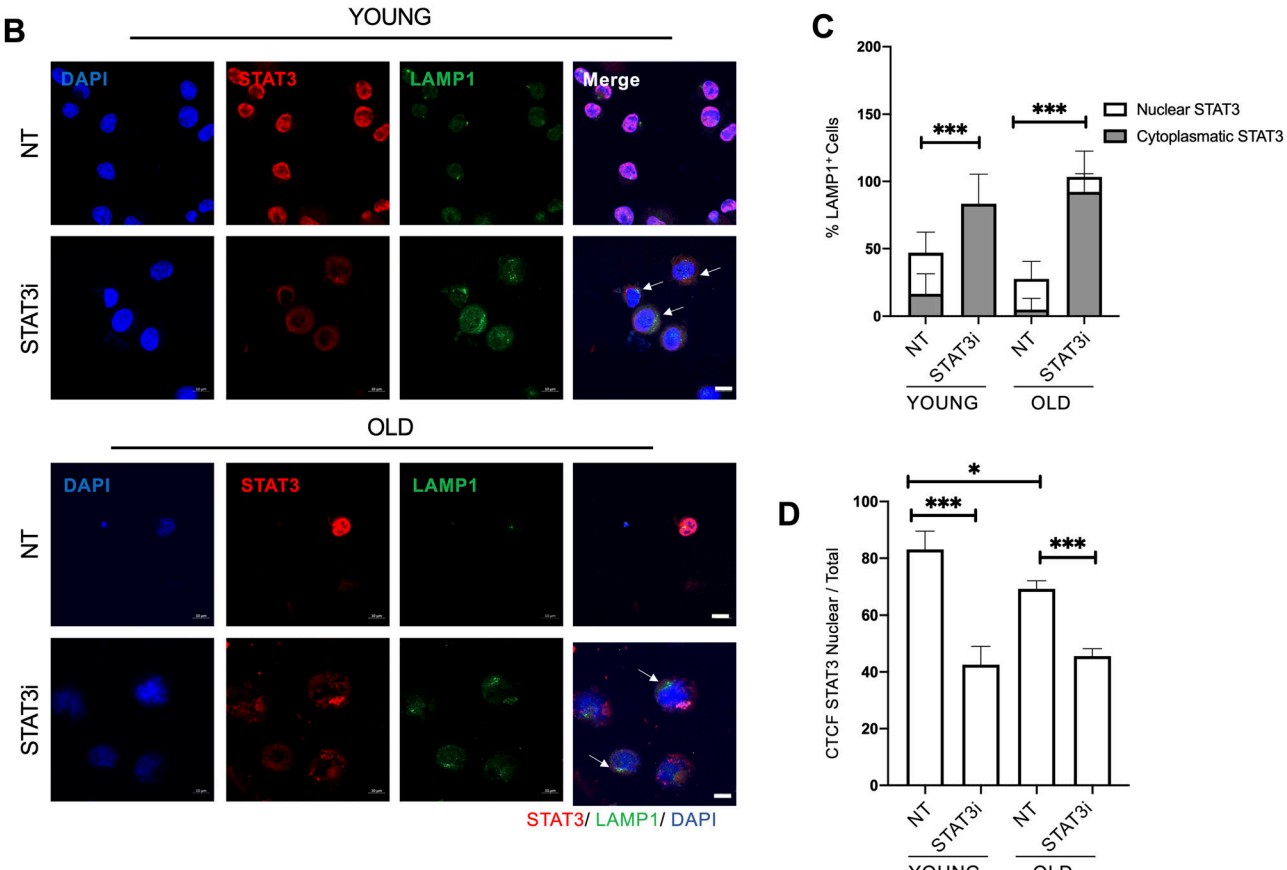

**Figure 5. STAT3 cytoplasmic translocation upon STAT3i treatment is coupled with the activation of autophagy in young and old muscle stem cells (MuSCs).**
**(A)** Schematic representation of MuSC isolation from 2 (young)- or 15 (old)-month-old WT mice not treated (NT) or treated with the STAT3i. MuSCs were isolated 3 d.p.i.
**(B)** Representative images of LAMP1 (green), STAT3 (red), and DAPI (blue) immunostaining in young or old MuSCs NT or treated with the STAT3i. Arrows represent cells with cytoplasmic retention of STAT3 and active autophagy. Scale bar = 10 $\mu m$. **(C)** Plot represents the percentage of LAMP1$^+$ MuSCs with a cytoplasmic versus nuclear STAT3 localization. (n = 3; 20 MuSCs per biological sample were counted; values represent the mean ± s.d., ***$P < 0.001$ by one-way ANOVA). **(D)** Plot represents the corrected total cell fluorescence of STAT3 nuclear over the total STAT3 (n = 3; 20 MuSCs per biological sample were counted; values represent the mean ± s.d., *$P < 0.05$ and ***$P < 0.001$ by one-way ANOVA).

Overall, we demonstrated that the STAT3–autophagy axis is perturbed in old MuSCs, strengthening the notion that the intrinsic age-associated clock in stem cells can be pharmacologically manipulated to prevent their functional decline.

# Materials and Methods

## Mouse model and muscle injury

Mouse lines used in this study were as follows: C57BL/6J (WT) mice were purchased from Jackson Laboratories. STAT3f/f, MCK-Cre; STAT3f/f, and Pax7-CreER;STAT3f/f mice were provided by Dr. A.

Sacco from the Sanford Burnham Prebys Medical Discovery Institute Animal Facility. C57BL/6J and Pax7-CreER;STAT3f/f mouse lines were analyzed at 2 or 15 mo of age. Muscle injury was performed by i.m. injection of 10 $\mu l$ of cardiotoxin (10 $\mu M$) into TA muscles. At indicated times, TA muscles were collected from the mice immediately after euthanasia. All animal procedures were approved by the Italian Ministry of Health and Istituto Superiore di Sanità (approval number 377/2018-PR).

## Cell line

C2C12 cells were cultured in growth medium, DMEM (Life Technologies), supplemented with high glucose, GlutaMAX, pyruvate,

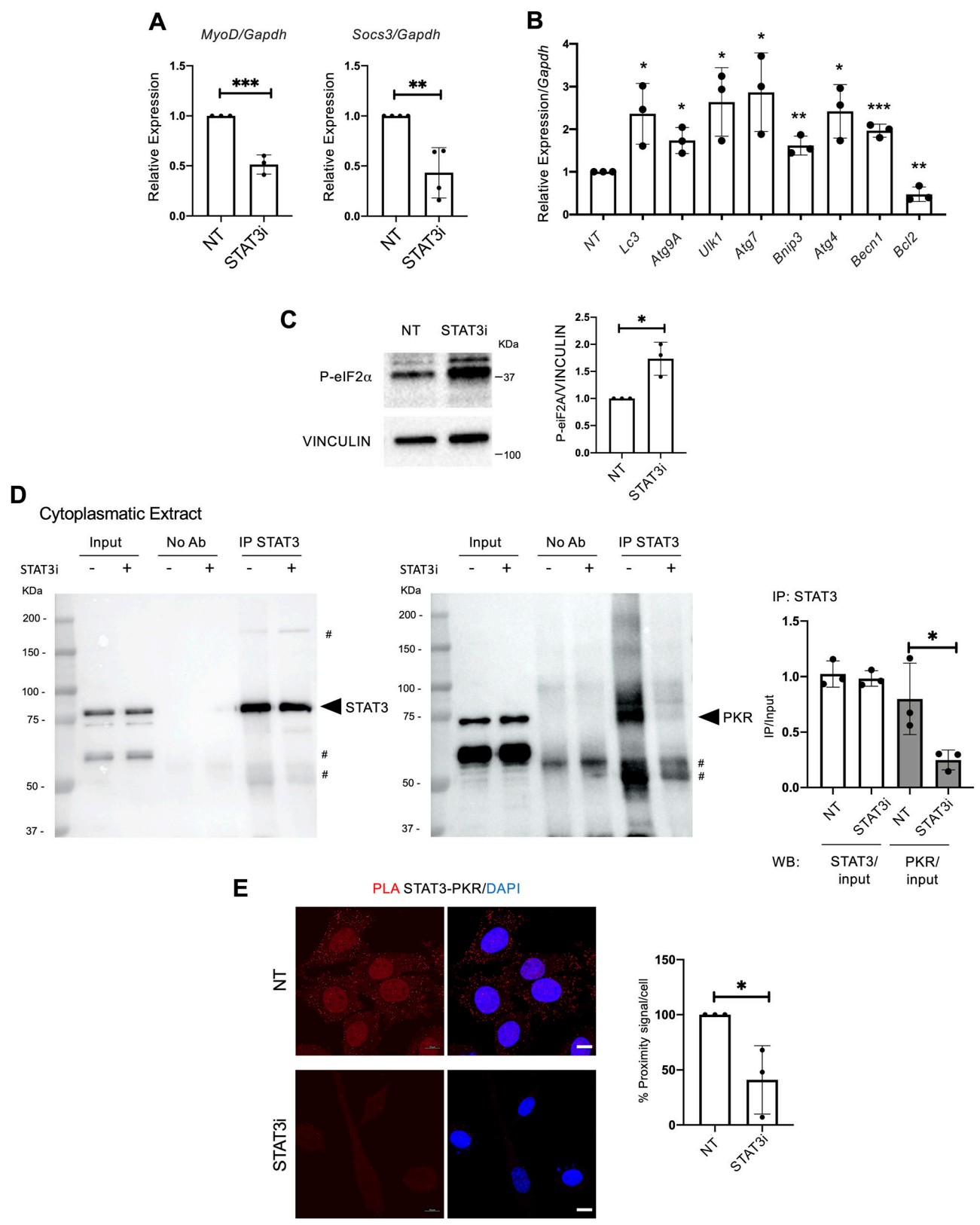

**Figure 6. STAT3 inhibition induces autophagy by disrupting the STAT3/PKR complex and activating pro-autophagic gene transcription in vitro.**
C2C12 cell line was treated for 24 h with PBS (NT) or the STAT3i at the concentration of 50 µM, then harvested for analysis. **(A)** qRT–PCR for *Myod* and *Socs3* normalized for *Gapdh* (n = 3 for *MyoD* and n = 4 for *Socs3*; values represent the mean ± s.d., **P < 0.01 and ***P < 0.001 by a t test). **(B)** qRT–PCR for *Lc3, Atg9A, Ulk1, Atg7, Bnip3, Atg4, Becn1,* and *Bcl2* normalized for *Gapdh* (n = 3; values represent the mean ± s.d., *P < 0.05, **P < 0.01, and ***P < 0.001 by a t test). **(C)** Phospho-eIF2α levels in NT or STAT3i-treated

10% FBS (Gibco), and penicillin/streptomycin at 37°C in a humidified atmosphere of 5% $CO_2$.

## STAT3i treatment

For in vitro studies, 50 μM STAT3i (#573096 from Sigma-Aldrich) was supplemented in growth medium. For in vivo studies in WT mice, 1 d after CTX injury, the STAT3i was injected i.m. at the dosage of 50 μg. Tissues were harvested after 3 and 5 d.p.i. for WB, RNA, and histological analyses.

## Tamoxifen treatment

2- and 15-mo-old Pax7-CreER;STAT3f/f mice were treated with tamoxifen (Cat #T5648; Sigma-Aldrich), 3 mg, suspended in corn oil, and injected intraperitoneally (i.p.) each day for 5 d. After 7 d from the last injection, TA muscles were injured with 10 μl of CTX i.m. Tissues were harvested after 3 d.p.i. for RNA and WB analyses and 5 d.p.i. for histological analysis.

## Autophagic flux assessment

For in vivo studies on total muscle, mice were treated with i.p. injections of autophagic lysosomotropic agent CLQ (50 mg/kg in PBS; Sigma-Aldrich) 4 h before mouse euthanasia to assess the autophagic flux. To determine the autophagic flux on freshly isolated MuSCs, CLQ was administered as described by Muñoz-Cánoves's laboratory (Campanario et al, 2021). In brief, digested muscles were treated with CLQ or vehicle (PBS) for 4 h before MuSC isolation by FACS. For in vitro studies, C2C12 cells were treated with CLQ at the dosage of 30 μM 4 h before cell harvesting (Perry et al, 2009). Autophagy flux was determined as the relative change in the LC3II/GAPDH ratio between CLQ-treated and not treated samples (ΔLC3II/GAPDH).

## Immunofluorescence

Cells and cryosections were fixed in PFA 4% (Sigma-Aldrich) and permeabilized with 100% methanol at −20°C or Triton 0.25%. The primary antibody immunostaining was performed O/N at 4°C. The following antibodies were used: LC3 (1:100, #2775S; Cell Signaling Technology), LAMP1 (1:200, #ab24170; Abcam), LC3 (1:200, #M152-3; MBL Int.), STAT3 (1:100, #119352; Abcam), P-STAT3$^{Y705}$ (1:1,000, #9145; mAb), PKR (1:100, #184257; Abcam), laminin (1:400, #L9393; Sigma-Aldrich), PAX7 (1:20; Developmental Studies Hybridoma Bank, DSHB), and embryonal myosin heavy chain (1:10; Developmental Studies Hybridoma Bank, DSHB). Alexa Fluor secondary antibodies (Molecular Probes) were used, and nuclei were counterstained with DAPI. EdU incorporation was revealed using the "Click-iT EdU Cell Proliferation Kit for Imaging, Alexa Fluor 594 dye" (Cat #C10354;

Thermo Fisher Scientific) following the manufacturer's protocol. PLA analysis for STAT3 and PKR was performed using Duolink PLA (In Situ Detection Reagents Red, #DUO92008; Duolink, In Situ PLA Probe Anti-Rabbit PLUS, #DUO92002; Duolink, In Situ PLA Probe Anti-Minus, #DUO92004; Duolink) following the manufacturer's protocol. PLA was expressed as % of proximity signal/cell and was estimated by ImageJ software. In MuSCs, PLA was expressed as the number of dots/cell.

The transverse sections and cultured cells were visualized on a Zeiss confocal microscope. The cells positive for the staining described in the text were quantified using ImageJ software. The LAMP1/LC3 co-localization was calculated by ImageJ software. The cross-sectional area was also calculated using ImageJ software and the Macro seg 5 modif.ijm specific plugin.

All histological analyses were performed blind to the treatment group by an investigator.

The figures reported are representative of all the examined fields.

## Histological analysis

For hematoxylin and eosin staining, frozen muscle sections were washed with PBS to allow rehydration and fixed in 4% PFA for 10 min. Then, sections were stained in hematoxylin (100%; Sigma-Aldrich) for 8 min, washed in running tap water, and then counterstained with eosin (100%; Sigma-Aldrich) for 1 min. After rinsing in distilled water, cryosections were dehydrated with increasing percentages of ethanol (Sigma-Aldrich), fixed in o-Xylene (Sigma-Aldrich), and mounted with Eukitt medium (Sigma-Aldrich).

## Cell preparation and isolation by FACS

Tibialis anterior muscles were subjected to enzymatic dissociation in PBS (Cat #14040-091; Gibco) with 2 mg/ml collagenase A (Cat # 10 103 586 001; Roche), 2.4 U/ml Dispase I (Cat # 04 942 078 001; Roche), 10 ng/ml DNase (Cat #11 284 932 001; Sigma-Aldrich), 0.4 mM $CaCl_2$, and 5 mM $MgCl_2$ for 60 min at 37°C under gentle agitation. The supernatants were filtered through a 100- and 40-μm cell strainers (#08-771-19, #08-771-1; BD Falcon) and incubated with the following antibodies for 30 min on ice: CD45-eFluor 450 (1/50, #48-0451-82; eBioscience), CD31-eFluor 450 (1/50, #48-0311-82; eBioscience), TER-119-eFluor 450 (1/50, #48-5921-82; eBioscience), Sca1-FITC (1/50, Ly-6A/E FITC, clone D7, #11-5981-82; eBioscience), Itga7-649 (1/500, #67-0010-01; AbLab). MuSCs were isolated as TER-119−/CD45−/CD31−/Itga7+/SCA-1− cells. Isolated satellite cells were used either for RNA extraction and WB analysis, or plated on glass slides (177402; Thermo Fisher Scientific) for immunostaining analysis.

---

C2C12. The plot represents the P-eIF2α/vinculin ratio (n = 3; values represent the mean ± s.d., *P < 0.05 by a t test). **(D)** Cytoplasmic extracts were immunoprecipitated for STAT3 and decorated for STAT3 (left panel) and PKR (right panel) to assess the STAT3/PKR complex (# aspecific bands). The plot represents the levels of immunoprecipitation (IP) over the input (n = 3; values represent the mean ± s.d., *P < 0.05 by a t test). **(E)** Proximity ligation assay (Alexa Fluor 594 dye, red) for STAT3 and PKR. Quantification of the percentage of the proximity signal for each nucleus in NT or STAT3i-treated cells. (n = 3; 30 muscle stem cells per biological sample were counted; values represent the mean ± s.d., *P < 0.05 by a t test).

## Protein extraction and Western blotting

The total protein extract was obtained by homogenizing primary myoblasts or sorted cells in RIPA buffer added with protease and phosphatase inhibitors. Protein extracts from muscles were obtained by tissue homogenization with TissueRuptor in lysis buffer (320 mM sucrose, 50 mM NaCl, 50 mM Tris, pH 7.5, 10% glycerol, 1% Triton). After being sonicated, cell debris was removed by centrifugation. Protein extracts were loaded in SDS gel and transferred to PVDF or nitrocellulose membranes. The following antibodies were used: LC3B (#2775S; Cell Signaling), p62 (#P0067; Sigma-Aldrich), total STAT3 (#4904S; Cell Signaling), P-STAT3 (#4113S; Cell Signaling), MYOD (5.8; Novus Biologicals), phospho-eIF2$\alpha$ (#3398S; Cell Signaling), GAPDH (Santa Cruz), and vinculin (sc-73614; Santa Cruz). HRP-conjugated secondary antibodies were revealed with the ECL chemiluminescence kit (Thermo Fisher Scientific).

## Co-immunoprecipitation

C2C12 cells were subjected to cell lysis buffer containing 10 mM Tris–HCl (pH 8), 10 mM NaCl, 0,1 mM EDTA (pH 8), 0,1 mM EGTA, and phosphatase and protease inhibitors. Then, NP-40 was added at the final concentration of 0.5%. After the centrifugation, 200 $\mu$g of cytoplasmic extract was immunoprecipitated with 2 $\mu$l of anti-STAT3 antibody (#119352; Abcam) overnight at 4°C. After the addition of 20 $\mu$l of protein A/G magnetic beads (Thermo Fisher Scientific) for 120 min at 4°C, the immunoprecipitated proteins were washed with lysis buffer, containing 50 mM Tris–HCl (pH 8), 150 mM NaCl, 1 mM EDTA (pH 8), 1 mM EGTA, six times. Subsequently, immunoprecipitated samples were added to Laemmli buffer and subjected to immunoblotting analysis. Incubation with primary and secondary antibodies was performed. The following antibodies have been used: STAT3 (79D7) (1:1,000, #4904; Cell Signaling), PKR (1:1,000, #184257; Abcam).

## RNA analysis by quantitative PCR

Total RNA was extracted with TRIzol, quantified with a NanoDrop 8000 spectrophotometer (Thermo Fisher Scientific), and retro-transcribed using the TaqMan reverse transcription kit (Applied Biosystems). The generated cDNA was used as a template in real-time PCRs with LightCycler 480 SYBR Green 1 Master Mix (Cat #TQ1211; SMOBIO) and was run on a Roche LC480 machine using three-step amplification and melt curve analysis. Quantitative real-time PCRs consisted of 2× SYBR Green Supermix, 0.25 mmol l$^{-1}$ forward and reverse primers (listed below), and 10 ng cDNA. Relative expression values were normalized to the housekeeping gene *Gapdh*; expression value was calculated using the $2^{-\delta\Delta CT}$ method. The sequences of oligonucleotides employed are as follows: *Atg4* (For: CCAGCTATTGATTGGAGGTGGA; Rev: CCAACTCCCATTTGCGCTATC), *Atg7* (For: CACAGTGGTGAGGCCAACTC; Rev: ACTGTTCTTACCAGCCT-CACTG), *Atg9A* (For: TTCGGATCCCAATGTCTGCC; Rev: GATGCTCTTTCTG CGTCTGC), *Becn1* (For: AGCCTCTGAAACTGGACACG; Rev: GCTGTGGTAA GTAATGGAGCTGT), *Bcl2* (For: AGCATGCGACCTCTGTTTGA; Rev: GCCA-CACGTTTCTTGGCAAT), *Bnip3* (For: TTCCCTAGCACCTTCTGATGA; Rev: GAACACCGCATTTACAGAACAA), *Gapdh* (For: TGCACCACCAACTGCTTAG; Rev: GGATGCAGGGATGATGTTC), *LC3* (For: CACTGCTCTGTCTTGTGTAGG

TTG; Rev:TCGTTGTGCCTTTATTAGTGCATC), *Myod1* (For: GGCTACGA-CACCGCCTACTA; Rev: CGACTCTGGTGGTGCATCTG), *p62* (For: ACTGCTCA GGAGGAGACGAT-3 Rev: CCGGGGATCAGCCTCTGTAG), *Stat3* (For: TGAAGGTGGTGGAGAACCTC; Rev: GCTGCTGCATCTTCTGTCTG); *Socs3* (For: TGCAGGAGAGCGGATTCTAC; Rev: TGACGCTCAACGTGAAGAAG), *Ulk1* (For: TTACCAGCGCATCGAGCA; Rev: TGGGGAGAAGGTGTGTAGGG).

## Single myofiber isolation

Single fibers were isolated from EDL muscle of MCK-Cre;STAT3f/f and STAT3f/f mice as described previously (Pasut et al, 2013; Gallot et al, 2016) and cultured in growth medium (GM: DMEM + pyruvate + 4.5*g*/liter glucose + glutamate, 10% horse serum (HS), 0.5% chicken embryo extract) for 4 h in the presence or not of CLQ to assess the autophagic flux.

For time course analysis, single fibers were isolated from EDL muscle of C57BL/6J WT mice and Pax7-CreER;STAT3f/f and cultured in proliferating medium GM1 (Tucciarone et al, 2018) (GM1: DMEM + pyruvate + 4.5*g*/liter glucose + glutamate, 10% horse serum (HS), 0.5% chicken embryo extract) for 24 h. Then, myofibers were exposed to GM2 medium (GM2: DMEM + pyruvate + 4.5*g*/liter glucose + glutamate, 20% FBS, 10% horse serum (HS), 1% chicken embryo extract) and STAT3 inhibition 50 $\mu$M until myofiber harvesting. Cell proliferation was measured by EdU incorporation, 4-h pulse.

## Statistical analysis and reproducibility

Data are presented as the mean ± SD of at least three independent biological replicates. The number of independent experimental replications is reported (n).

Statistical analysis was conducted using Prism 7.0 A software (Pad Software). All data meet the assumptions of the tests (e.g., normal distribution). Statistical significance was determined using an unpaired, two-tailed *t* test or a two-sided Mann–Whitney U test to compare the means of two groups, whereas one-way ANOVA was used for comparison among more than two groups. Tukey's test was used for multiple comparison analysis.

Statistical significance was defined as $P < 0.05$ (*), $P < 0.01$ (**), and $P < 0.001$ (***).

Immunofluorescence images are representative of at least three different experiments.

# Supplementary Information

# Acknowledgements

L Latella is supported by the Italian Ministry of Health no. PE-2016-02363049 and AFM no. 24406. This project has received funding from the European Union's Horizon 2020 research and innovation program under the Marie Skłodowska-Curie grant agreement no. 860034 to L Latella; PL Puri is supported by R01AR076247-01 and R01 AR056712; A Sacco is supported by NIH R01 AR077448.

## Author Contributions

G Catarinella: data curation, formal analysis, validation, and investigation.
A Bracaglia: data curation, formal analysis, validation, and investigation.
E Skafida: data curation, formal analysis, validation, and investigation.
P Procopio: data curation, formal analysis, validation, and investigation.
V Ruggieri: formal analysis.
C Parisi: formal analysis.
M De Bardi: methodology.
G Borsellino: methodology.
L Madaro: formal analysis.
PL Puri: conceptualization, data curation, funding acquisition, and writing—original draft, review, and editing.
A Sacco: conceptualization, data curation, formal analysis, supervision, funding acquisition, project administration, and writing—original draft, review, and editing.
L Latella: conceptualization, data curation, formal analysis, supervision, funding acquisition, project administration, and writing—original draft, review, and editing.

## Conflict of Interest Statement

The authors declare that they have no conflict of interest.

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
