## [Reviewer comments · Life Science Alliance]

Life Science Alliance

STAT3 inhibition recovers regeneration of aged muscles by restoring autophagy in muscle stem cells

Giorgia Catarinella, Andrea Bracaglia, Emilia Skafida, Paola Procopio, Veronica Ruggieri, Cristina Parisi, Marco De Bardi, Giovanna Borsellino, Luca Madaro, Pier Puri, Alessandra Sacco, and Lucia Latella

DOI: <https://doi.org/10.26508/lsa.202302503>

Corresponding author(s): Lucia Latella, National Research Council

Review Timeline:

Submission Date:	2023-11-29
Editorial Decision:	2024-01-19
Revision Received:	2024-05-21
Editorial Decision:	2024-05-23
Revision Received:	2024-05-24
Accepted:	2024-05-24

Transaction Report:

January 19, 2024

Re: Life Science Alliance manuscript #LSA-2023-02503-T

Dr. Lucia LATELLA
National Research Council
Institute of Translational Pharmacology
Via del Fosso di Fiorano 64
Via Fosso del Cavaliere 100
Rome, Italy 00143
Italy

Dear Dr. Latella,

Thank you for submitting your manuscript entitled "STAT3-mediated autophagy restores MuSC-mediated regeneration by transcription-dependent and independent mechanisms" to Life Science Alliance. The manuscript was assessed by expert reviewers, whose comments are appended to this letter. We invite you to submit a revised manuscript addressing the Reviewer comments.

Thank you for this interesting contribution to Life Science Alliance. We are looking forward to receiving your revised manuscript.

Sincerely,

B. MANUSCRIPT ORGANIZATION AND FORMATTING:

Reviewer #1 (Comments to the Authors (Required)):

In this manuscript, Catarinella et al. show that STAT3 inhibition or deletion induces autophagy in young and aged skeletal muscles during regeneration and drives MuSCs expansion and differentiation. STAT3 inhibition triggers the disruption of STAT3/PKR cytoplasmic interaction leading to phosphorylation of eIF2alpha, which has been linked to increased autophagy. Overall, the findings are interesting and the experiments are well conducted. Nonetheless, certain data necessitate further validation to ensure the accuracy of the conclusions drawn.

Figures 1 A and B. GAPDH is not a satisfactory normalization protein for these blots. Especially in Fig 1B, its levels are very much increased by CLQ treatment, significantly affecting the DeltaLC3II/GAPDH quantification. A more reliable housekeeping protein should be used.

Figure 3C. p62 protein levels are affected by degradation, as the authors highlighted, but also by transcription modulation. Here, the mRNA levels of p62 with or w/o STAT3i should be measured.

Figure 2C. Here, the effects of CLQ on LC3II accumulation on NT samples are not as apparent as they should be. CLQ concentration and treatment time to get LC3-II accumulation should be carefully assessed in control samples (NT), as positive control.

Figure 3 B-C, 4C, 5B. LAMP1 staining is not sufficient to demonstrate autophagy induction. LC3 immunofluorescence and LAMP1-LC3 colocalization should be assessed.

Figure 4B. Left panel (young). The effects of CLQ on LC3-II accumulation are not apparent: the experimental setup should be carefully revised, as suggested in fig. 2C.

Figure 4B. Right panel (old). Why does CLQ treatment reduce LC3 levels (both LC3-I and -II) in NT samples? GAPDH is also reduced: does CLQ treatment reduce MuSC viability?

Figure 6. Increased Becn1 mRNA levels are not sufficient to claim increased autophagy gene expression. RT-PCR for additional autophagy-related genes should be performed.

Reviewer #2 (Comments to the Authors (Required)):

In this study, Catarinella and colleagues investigated the impact of STAT3 on the regulation of the autophagic flux in MuSC in young and old mouse. Their findings identify a new mechanism regulating MuSC myogenic progression. While these findings are of interest, there is additional experiments and controls needed to confirm some of the findings.

Main comments:

- In figure 1 the authors use specific STAT3 inhibitor, but they do not demonstrate that this compound is efficient and specific at the concentration and conditions used. The addition of controls measuring the impact on STAT3 and other STAT isoforms is needed.
- It is unclear why the authors used 15-month old mice. Many studies have shown that MuSC defects appears at older age in mice (22-24 months and older). In their previous Nature Medicine paper (2014), the authors used 24-months old mice. Can the authors show that STAT3 is actually upregulated in 15-months old mice.
- Figure 2D shows the results of Myfiber CSA at 5 dpi. Please add representative micrographs. Moreover, a more complete analysis using different markers (MyHC-embryonic, number of nuclei/fiber, etc) and/or time point would be more convincing.
- In the figure 3, the authors used LAMP1 as a marker of autophagy. This marker is used as a marker of lysosomal content or morphology, but on its own it is not the best marker to assess autophagy (see Guidelines for the use and interpretation of assays for monitoring autophagy: "In addition, structural proteins of the lysosome/late endo-somes, such as LAMP1 and LAMP2 or SCARB2/LIMP-2, can be used for confirmation. No single protein marker, however, has been effective in discriminating autolysosomes from the compartments mentioned above, in part due to the dynamic fusion and "kiss-and-run" events that promote interchange of components that can occur between these organelle subtypes. Rigorous further discrimination of these compartments from each other and other vesicles ultimately requires demonstrating the colocalization of a second marker indicating the pre-sense of an autophagic substrate (e.g., LC3 and CTSD [cathepsin D] colocalization) or the acidification of the compartment (e.g., mRFP/mCherry-GFP-LC3 probes or LysoTracker{trade mark, serif} dyes; see Tandem mRFP/mCherry-GFP fluorescence microscopy), Keima probes, or BODIPY-pepstatin A that allows detection of CTSD in an activated form within an

acidic compartment), and, when appropriate, by excluding markers of other vesicular components [76, 80, 81]").

- In suppl. fig. 3A, the authors showed a drastic effect of STAT3i on young muscle regeneration at 5 dpi (2-fold increase in CSA). However, the micrographs show that many fibers in the STAT3i group are not centronucleated, suggesting that they have not been appropriately injured. There is a lot of heterogeneity in the different parts of the muscle at 5 dpi (especially that the authors injected only 10ul of CTX, which might not be enough to induce homogenous injury). How many myofibers were counted? I would highly recommend assessing the whole muscle section, which is easy nowadays with specific plugins. To this end, please add a reference for the plugin used (Macro seg 5).
- It is unclear why the authors suddenly decided to use C2C12 cells for the last experiments. What is the rationale?

Minor comments

- References 11 and 35 are the same.
- Add reference to support the sentence : "On the other hand, cytoplasmic STAT3 constitutively represses autophagy by sequestering protein kinase R (PKR) and preventing the phosphorylation of eukaryotic initiation factor 2 α (eIF2 α)."
- Please add a supplemental figure including all the uncropped and unedited blots.
- Please indicate in the figure legends the minimum number of myofibers or cells counted per biological samples.
- In the PLA experiment, it is unclear the meaning of %area/nucleus. Please provide additional details.

Reviewer #3 (Comments to the Authors (Required)):

The current study by Catarinella et al investigates the role of STAT3 in the regulation of autophagy in muscle stem cells (MuSCs) based on inhibition of PKR, a critical kinase required for phosphorylation of EIF2A, a known autophagy activator. The authors describe that inhibition of STAT3 in MuSCs stimulates autophagic flux, resulting in expansion of MuSCs and improvement of skeletal muscle regeneration. Experiments in the C2C12 myoblast cell line indicate that inhibition of STAT3 disrupts the STAT3-PKR complex and increases phosphorylation of EIF2A, thereby promoting initiation of autophagy.

The role of STAT3 in regulating and suppressing expansion of MuSCs has already been studied extensively but the potential function of cytoplasmic STAT3 for inhibiting autophagy has not been analyzed in detail, albeit the underlying basic mechanism was already described more than 10 years ago. In principle, the study is well done and controlled but does not go very deep. New mechanistic insights, e.g. regarding the still rather enigmatic role of EIF2A phosphorylation for the regulation of autophagy, are not provided. In light of the potential therapeutic applications of STAT3 inhibitors for improving muscle regeneration in aged individual this may be acceptable. However, I noted a major conceptual problem and other shortcomings that need to be addressed.

Specific comments

The authors argue that enhanced continuous JAK3/STAT3 signaling in "inflammaging" muscles inhibits autophagy in MuSC, preventing proper MuSC expansion for muscle regeneration. According to the authors, inhibition of STAT3 disrupts this vicious cycle, restores autophagy, and improves muscle regeneration in aged mice. At the first glance, such a chain of arguments sounds compelling but I do have problems to reconcile this hypothesis with the cytoplasmic inhibition of PKR by STAT3. According to the literature, both wildtype STAT3 and nonphosphorylatable STAT3Y705F inhibit baseline as well as starvation-induced EIF2A phosphorylation. With other words, STAT3 inhibits autophagy without JAK2-induced phosphorylation. At least, that is how I understand the literature. Please correct me if I am wrong. To explain STAT3-dependent inhibition of autophagy in MuSCs the authors have to analyze the cytoplasmic levels of STAT3 in young and aged MuSCs. Enhanced phosphorylation of STAT3 is most likely irrelevant for increased inhibition of autophagy in aged MuSCs. Just the opposite, enhanced translocation of phosphorylated STAT3 into the nucleus should attenuate STAT3-PKR dependent inhibition of autophagy. An increase of cytoplasmic STAT3 in aged MuSC would support the hypothesis of STAT3-PKR driven inhibition of autophagy in aged MuSCs and should be investigated.

The authors report an increase of PAX7-positive MuSCs shortly after inactivation of Stat3 in MuSCs of 2-months-old mice, confirming previous reports. In my opinion, it is critical to demonstrate such an increase of MuSCs after Stat3 inactivation in aged mice and also analyze the impact of a long-term loss of Stat3 on MuSC numbers in aged mice.

The authors claim that STAT3i treatment of old MuSCs restores autophagy to 'basal levels'. However, according to Fig. 4B STAT3i treated old MuSCs appear to have a lower ratio of LC3II/GAPDH than young non-treated MuSC. If there is no significant difference, please indicate that in the figure.

For the mechanistic studies, the authors used the C2C12 myoblast cell line, which has very different properties than primary MuSCs. The authors have to repeat at least one critical experiment in primary MuSCs to make sure that C2C12 cells are an appropriate model to study STAT3-dependent inhibition of autophagy via PKR and EIF2A.

To characterize transcriptional effects of STAT3 inhibition, the authors performed some RT-PCR assays, which are not very informative. Why not a RNAseq of C2C12 AND primary MuSCs treated with STAT3i? Similar transcriptional effects of STAT3i in C2C12 and primary MuSCs would also justify the use of C2C12 cells as an adequate model for the mechanistic STAT3i experiments.

Wildtype control mice are missing in the experiments depicted in Fig. 4E. How can the authors claim 'that STAT3 deletion is sufficient to activate the autophagic flux in MuSCs irrespective of whether they were isolated from young or old mice' without comparing to MuSCs that express Stat3?

The authors claim improved skeletal muscle regeneration of young and aged mice upon STAT3i treatment. The claim is exclusively based on quantification of cross-sectional areas at a single time point after muscle damage, which is not acceptable. The authors have to assess different time points after damage and also stain for embryonic myosin heavy chain as an early marker for regenerating fibers. H&E staining should be supplied as well. Actually, it was not clear from the description when the analysis was done. The authors wrote that muscles were analyzed 5 days after injection of STAT3i. Was the muscle damage done before injecting STAT3i? The extent of muscle regeneration suggests that muscle injury was induced about 10 days before analysis.

The co-IP shown in Fig. 6B lacks proper quantification, which needs to be added. It would also be helpful to immunoprecipitate PKR and probe with STAT3 antibodies to validate the effects of STAT3i on the interaction between PKR and STAT3. Molecular weight markers are missing for Fig. 6B. What do the different bands on the PKR blot represent?

Rome 20/05/2024

Lucia Latella, PhD
Institute of Translational Pharmacology,
National Research Council
Via del Fosso di Cavaliere, 100
Roma 00133, Italy
e-mail: l.latella@hsantalucia.it
Lucia.Latella@ift.cnr.it

Dear Dr. Sawey,

We were glad to read the positive comments from the referees, who expressed unanimously interest for our manuscript (#LSA-2023-02503-T).

They provided some suggestions/comments that we have addressed, as described in the point-to-point rebuttal below.

Reviewer #1 (Comments to the Authors (Required)): -

In this manuscript, Catarinella et al. show that STAT3 inhibition or deletion induces autophagy in young and aged skeletal muscles during regeneration and drives MuSCs expansion and differentiation. STAT3 inhibition triggers the disruption of STAT3/PKR cytoplasmic interaction leading to phosphorylation of eIF2alpha, which has been linked to increased autophagy.

Overall, the findings are interesting and the experiments are well conducted. Nonetheless, certain data necessitate further validation to ensure the accuracy of the conclusions drawn.

1. Figures 1 A and B. GAPDH is not a satisfactory normalization protein for these blots. Especially in Fig 1B, its levels are very much increased by CLQ treatment, significantly affecting the DeltaLC3II/GAPDH quantification. A more reliable housekeeping protein should be used.

RE: We thank the reviewer for the comment. Indeed, while executing our analyses on the autophagic flux, we did not observe a correlation among CLQ treatment and GAPDH levels. However, according to the reviewer suggestion, we ran a set of experiments that was normalized for both GAPDH and beta-actin. During the execution of these experiments, we were able to determine that two independent normalizer genes showed the same trend as shown in the figure below (only for the reviewer). Therefore, we can consider GAPDH a reliable housekeeping protein for normalization in our model. Nevertheless, we have replaced the image from Figure 1B with a new representative blot (New Figure 1C).

[Figure removed by editorial staff per authors' request]

2. Figure 3C-1C. p62 protein levels are affected by degradation, as the authors highlighted, but also by transcription modulation. Here, the mRNA levels of p62 with or w/o STAT3i should be measured.

RE: We thank the reviewer for this suggestion. The measurement of p62 mRNA levels as a marker of autophagic flux is still controversial and can be misinterpreted mainly because this protein is subject to complex regulation at both the transcriptional and post-translational levels. Yet, we now measured the transcriptional modulation of p62 upon STAT3 inhibition. Our data reveal an upregulation of p62 mRNA levels in both young (New Figure 1 E) and old (New Figure 1 H) mice treated with STAT3i. This might be explained as an adaptive response that, in conditions of active autophagy, tends to restore the expression of the autophagy substrate SQSTM1/p62 (Sahani et al, 2014). Moreover, it has been previously shown that the eIF2 α /ATF4 pathway directs the autophagy gene transcriptional program, including the upregulation of autophagic genes in response to amino acid starvation or endoplasmic reticulum stress. and be dependent on the eIF2 α /ATF4 pathway that directs the autophagy gene transcriptional program in response to amino acid starvation or ER stress (B'chir et al 2013).

3. Figure 2C. Here, the effects of CLQ on LC3II accumulation on NT samples are not as apparent as they should be. CLQ concentration and treatment time to get LC3-II accumulation should be carefully assessed in control samples (NT), as positive control.

RE: The autophagic activity is usually low under basal conditions, but can be markedly increased by stimuli such as nutrient deprivation, hypoxia, cellular stress and muscle regeneration. In Figure 2C, we measured the autophagic flux in uninjured muscles as relative change of the LC3II/GAPDH ratio between CLQ-treated and not treated samples (Δ LC3II/GAPDH). Our measurements range from 0.5 to 1.7 indicating a modest autophagic flux in control samples (NT). Given the basal autophagic flux in NT sample, we wanted to measure the impact of STAT3 inhibition on the autophagic flux. Our data show that STAT3 ablation in myofibers is not able to activate the autophagic flux.

4. Figure 3 B-C, 4C, 5B. LAMP1 staining is not sufficient to demonstrate autophagy induction. LC3 immunofluorescence and LAMP1-LC3 colocalization should be assessed.

RE: We agree with the reviewer on this observation. We ran additional immunostainings employing LC3 to detect autophagy on fixed samples. We performed a co-immunofluorescence of PAX7/LC3 on myofibers freshly isolated from mice *ex vivo* and treated with the STAT3 inhibitor (New Figure 3F, G). To perform this staining, we had to fix cells with Methanol/acetone that is not the recommended protocol to obtain good staining for PAX7. However, we managed to demonstrate significant co-localization of LC3 in PAX7 positive cells, strengthening the data already shown using LAMP1 as an autophagy marker. Likewise, we performed a co-immunostaining for LC3 and LAMP1 in MuSCs with or without STAT3i treatment demonstrating a clear co-localization of the two proteins (New Figure 4E). These data further support the notion that STAT3 is a key regulator of autophagy during MuSC activation.

5. Figure 4B. Left panel (young). The effects of CLQ on LC3-II accumulation are not apparent: the experimental setup should be carefully revised, as suggested in fig. 2C.

RE: We thank the reviewer for this observation. Indeed, we performed additional experiments on MuSC freshly isolated from young mice and analyzed immediately after cell sorting. A new, representative WB is now shown in New Figure 4C (left panel).

6. Figure 4B. Right panel (old). Why does CLQ treatment reduce LC3 levels (both LC3-I and -II) in NT samples? GAPDH is also reduced: does CLQ treatment reduce MuSC viability?

RE: During our analyses, we did not observe a relationship between the exposure to CLQ and the reduction of LC3I and LC3II, nor an effect of CLQ on MuSC viability. In New Figure 4C (right panel), the reduction of LC3 levels upon CLQ incubation in NT samples is only a matter of loading proteins in the gel. Yet, the autophagic flux is calculated as relative change of the LC3II/GAPDH ratio between CLQ-treated and not treated samples (Δ LC3II/GAPDH), therefore is normalized for protein content.

7. Figure 6. Increased *Becn1* mRNA levels are not sufficient to claim increased autophagy gene expression. RT-PCR for additional autophagy-related genes should be performed.

RE: Following the reviewer's suggestion, additional autophagy-related genes analyzed by q-RT-PCR are now included in New Figure 6B.

Reviewer #2 (Comments to the Authors (Required)):

In this study, Catarinella and colleagues investigated the impact of STAT3 on the regulation of the autophagic flux in MuSC in young and old mouse. Their findings identify a new mechanism regulating MuSC myogenic progression. While these findings are of interest, there is additional experiments and controls needed to confirm some of the findings.

Main comments:

1. In figure 1 the authors use specific STAT3 inhibitor, but they do not demonstrate that this compound is efficient and specific at the concentration and conditions used. The addition of controls measuring the impact on STAT3 and other STAT isoforms is needed.

RE: The STAT3 inhibitor we have used in this manuscript is a cell-permeable peptide, analog of Stat3-SH2 domain-binding phosphopeptide, that acts as a selective, potent blocker of Stat3 activation. The STAT3i has been provided accordingly to the concentration and conditions used in Tierney et al. In supplementary Figure S4A, we show the impact of STAT3 inhibition in C2C12 measuring STAT3 levels and the phosphorylation in Tyrosine 705. In addition, we assessed the transcription of STAT3 downstream target such as *MyoD* and *Socs3* (New Figure 6A). In the revised version of the manuscript, we provided additional evidence supporting the effectiveness of the treatment *in vivo*, showing the effect of the STAT3i treatment on Tyr705 STAT3 phosphorylation (New Figure 1B). In addition, the revised version of this MS includes data on the efficacy of the STAT3i treatment in MuSC assessed by measuring the two STAT3 target genes, namely *Socs3* and *MyoD*, that are downregulated upon STAT3i administration (New Figure 4B).

2. It is unclear why the authors used 15-month old mice. Many studies have shown that MuSC defects appears at older age in mice (22-24 months and older). In their previous Nature Medicine paper (2014), the authors used 24-months old mice. Can the authors show that STAT3 is actually upregulated in 15-months old mice.

RE: Following the reviewer's suggestion, the revised MS now includes data on STAT3 levels in 2 and 15-months-old mice showing the upregulation of the protein in 15 months-old mice (New Figure 1B).

3. Figure 2D shows the results of Myfiber CSA at 5 dpi. Please add representative micrographs. Moreover, a more complete analysis using different markers (MyHC-embryonic, number of nuclei/fiber, etc) and/or time point would be more convincing.

RE: Our results show that STAT3 deletion in myofibers did not affect autophagy. We decided to remove the data on myofiber CSA at 5 dpi since a similar finding was already published (Swiderski K, et al. *Skeletal Muscle*. 2016). This is consistent with other studies, in which the MCK-Cre STAT3 mouse model did not exhibit differences (1) in skeletal muscle insulin sensitivity compared to controls, and was similarly impaired by high fat diet (White et al, *Molecular Metabolism* 2015); (2) in overload-mediated hypertrophy in mouse skeletal muscle (Perez-Schindler et al, *Am J Physiol Cell Physiol* 2017); (3) in skeletal muscle mitochondrial and physiological function (Dent et al, *J Appl Physiol* (1985) 2019).

4. In the figure 3, the authors used LAMP1 as a marker of autophagy. This marker is used as a marker of lysosomal content or morphology, but on its own it is not be the best marker to assess autophagy (see Guidelines for the use and interpretation of assays for monitoring autophagy: "In addition, structural proteins of the lysosome/late endosomes, such as LAMP1 and LAMP2 or SCARB2/LIMP-2, can be used for confirmation. No single protein marker, however, has been effective in discriminating autolysosomes from the compartments mentioned above, in part due to the dynamic fusion and "kiss-and-run" events that promote interchange of components that can occur between these organelle subtypes. Rigorous further discrimination of these compartments from each other and other vesicles ultimately requires demonstrating the colocalization of a second marker indicating the presence of an autophagic substrate (e.g., LC3 and CTSD [cathepsin D] colocalization) or the acidification of the compartment (e.g., mRFP/mCherry-GFP-LC3 probes or LysoTracker {trade mark, serif} dyes; see Tandem mRFP/mCherry-GFP fluorescence microscopy), Keima probes, or BODIPY-pepstatin A that allows detection of CTSD in an activated form within an acidic compartment), and, when appropriate, by excluding markers of other vesicular components [76, 80, 81]").

RE: We thank this reviewer for suggesting additional tools to assess autophagy. As we replied to reviewer #1, we performed additional immunostainings employing LC3 to detect autophagy on fixed samples. We performed a co-immunofluorescence of PAX7/LC3 on myofibers freshly isolated from mice *ex vivo* treated with the STAT3 inhibitor (New Figure 3F and G). To perform this staining, we had to fix cells with Methanol/acetone, that is not the recommended protocol to obtain good staining for PAX7. However, we managed to demonstrate the significative co-localization of LC3 in PAX7 positive cells, strengthening the data already shown using LAMP1 as an autophagy marker. Likewise, we performed a co-immunostaining for LC3 and LAMP1 in MuSCs with and without STAT3i treatment demonstrating a clear co-localization of the two proteins (New Figure 4E). These data not only complement with an additional marker, together with LAMP1, in the analysis of the autophagic process, but supports our hypothesis of STAT3-dependent autophagy during MuSC activation.

5. In suppl. fig. 3A, the authors showed a drastic effect of STAT3i on young muscle regeneration at 5 dpi (2-fold increase in CSA). However, the micrographs show that many fibers in the STAT3i group are not centronucleated, suggesting that they have not been appropriately injured. There is a lot of heterogeneity in the different parts of the muscle at 5 dpi (especially that the authors injected only 10ul of CTX, which might not be enough to induce homogenous injury). How many myofibers were counted? I would highly recommend assessing the whole muscle section, which is easy nowadays with specific plugins. To this end, please add a reference for the plugin used (Macro seg 5).

RE: While measuring the CSA showed in Supplementary Figure 3A (now Supplementary Figure S2B), we measured the injured area using the ImageJ software and Macro seg 5 modif.ijm specific plugin, as described in the Methods. However, in the revised version of the MS, we further included H&E (Supplementary Figure S2C) and embryonal Myosin Heavy Chain staining (Supplementary

Figure S2D) indicating that the muscles were properly and widely injured. Our data indicate that the STAT3 inhibitor supports muscle regeneration, as evaluated by improved muscle morphology, and increased of myofiber size. This is consistent with the data we previously published (Tierney et al. 2014).

6. It is unclear why the authors suddenly decided to use C2C12 cells for the last experiments. What is the rationale?

RE: The rationale behind the use of C2C12 myoblasts to perform the experiments showed in Figure 6 lies on the need of adequate material to perform Co-IP experiments with nuclear and cytoplasmic extracts. Beside the fact that MuSCs are FACS-sorted (thereby a huge number of mice would have been used to reach the necessary amount of proteins), MuSCs are cells characterized by a smaller cytoplasm, as compared to C2C12 myoblasts. The same applies to the PLA experiments, where we sought to measure the interaction between PKR and STAT3 in the cytoplasmic region. Nevertheless, we performed a PLA for STAT3 and PKR in freshly isolated MuSC *in vivo* treated with vehicle or STAT3i confirming the disruption of the STAT3/PKR complex upon treatment with STAT3i (New Supplementary Figure S5B).

Minor comments

- References 11 and 35 are the same.

RE: We corrected this error.

- Add reference to support the sentence : "On the other hand, cytoplasmic STAT3 constitutively represses autophagy by sequestering protein kinase R (PKR) and preventing the phosphorylation of eukaryotic initiation factor 2 α (eIF2 α)."

RE: We have now added the reference (Shen et al. 2012).

- Please add a supplemental figure including all the uncropped and unedited blots.

RE: We have now added a supplementary figure with the uncropped and unedited blots.

- Please indicate in the figure legends the minimum number of myofibers or cells counted per biological samples.

RE: We have now added these details in the figure legends.

- In the PLA experiment, it is unclear the meaning of %area/nucleus. Please provide additional details.

RE: We thank the reviewer for this suggestion. Indeed, in the revised version of the MS, PLA was expressed as % of Proximity signal/cell (New Figure 6E). In MuSCs, PLA was expressed as number of dots/cell (New Supplementary Figure S5B).

Reviewer #3 (Comments to the Authors (Required)):

The current study by Catarinella et al investigates the role of STAT3 in the regulation of autophagy in muscle stem cells (MuSCs) based on inhibition of PKR, a critical kinase required for phosphorylation of EIF2A, a known autophagy activator. The authors describe that inhibition of STAT3 in MuSCs stimulates autophagic flux, resulting in expansion of MuSCs and improvement of skeletal muscle regeneration. Experiments in the C2C12 myoblast cell line indicate that inhibition of STAT3 disrupts the STAT3-PKR complex and increases phosphorylation of EIF2A, thereby promoting initiation of autophagy.

The role of STAT3 in regulating and suppressing expansion of MuSCs has already been studied extensively but the potential function of cytoplasmic STAT3 for inhibiting autophagy has not been

analyzed in detail, albeit the underlying basic mechanism was already described more than 10 years ago. In principle, the study is well done and controlled but does not go very deep. New mechanistic insights, e.g. regarding the still rather enigmatic role of EIF2A phosphorylation for the regulation of autophagy, are not provided. In light of the potential therapeutic applications of STAT3 inhibitors for improving muscle regeneration in aged individual this may be acceptable. However, I noted a major conceptual problem and other shortcomings that need to be addressed.

Specific comments

1. The authors argue that enhanced continuous JAK3/STAT3 signaling in 'inflammaging' muscles inhibits autophagy in MuSC, preventing proper MuSC expansion for muscle regeneration. According to the authors, inhibition of STAT3 disrupts this vicious cycle, restores autophagy, and improves muscle regeneration in aged mice. At the first glance, such a chain of arguments sounds compelling but I do have problems to reconcile this hypothesis with the cytoplasmic inhibition of PKR by STAT3. According to the literature, both wildtype STAT3 and nonphosphorylatable STAT3Y705F inhibit baseline as well as starvation-induced EIF2A phosphorylation. With other words, STAT3 inhibits autophagy without JAK2-induced phosphorylation. At least, that is how I understand the literature. Please correct me if I am wrong. To explain STAT3-dependent inhibition of autophagy in MuSCs the authors have to analyze the cytoplasmic levels of STAT3 in young and aged MuSCs. Enhanced phosphorylation of STAT3 is most likely irrelevant for increased inhibition of autophagy in aged MuSCs. Just the opposite, enhanced translocation of phosphorylated STAT3 into the nucleus should attenuate STAT3-PKR dependent inhibition of autophagy. An increase of cytoplasmic STAT3 in aged MuSC would support the hypothesis of STAT3-PKR driven inhibition of autophagy in aged MuSCs and should be investigated.

RE: The role of STAT3 in modulating autophagy is described through several mechanisms, although not fully deciphered to date. According to the literature, Il-6-induced phosphorylation of STAT3 in Tyrosine 705 reduces autophagy (Qin B et al. Sci Rep. 2015.). Nonetheless, cytoplasmic non-phosphorylated STAT3, rather than p-STAT3, represses autophagy by sequestering PKR. Shen and colleagues have shown that cytoplasmic STAT3, rather than p-STAT3, participates in the regulation of autophagy. To explain STAT3-dependent inhibition of autophagy in MuSCs, as this reviewer suggests, we now show that the levels of cytoplasmic STAT3 in aged MuSCs are increased as compared with young MuSCs (New Figure 5D). Upon STAT3i treatment, the STAT3/PKR complex is disrupted leading to the recovery of the autophagic process in old MuSCs.

2. The authors report an increase of PAX7-positive MuSCs shortly after inactivation of Stat3 in MuSCs of 2-months-old mice, confirming previous reports. In my opinion, it is critical to demonstrate such an increase of MuSCs after Stat3 inactivation in aged mice and also analyze the impact of a long-term loss of Stat3 on MuSC numbers in aged mice.

RE: In the revised version of the MS, we included data reporting an increase of PAX7 in both 2- and 15-months-old mice (New Supplementary Figure S1).

3. The authors claim that STAT3i treatment of old MuSCs restores autophagy to 'basal levels'. However, according to Fig. 4B STAT3i treated old MuSCs appear to have a lower ratio of □LC3II/GAPDH than young non-treated MuSC. If there is no significant difference, please indicate that in the figure.

RE: We measured the significance of Delta LC3/GAPDH among old MuSC treated with STAT3i and young non-treated and we rephrase the sentence: "Upon STAT3i treatment, we detected an activation

of the autophagic flux in young (Fig. 4C – left panels) and old MuSCs (Fig. 4C – right panels) as measured by the increase in Δ LC3II/GAPDH.”

4. For the mechanistic studies, the authors used the C2C12 myoblast cell line, which has very different properties than primary MuSCs. The authors have to repeat at least one critical experiment in primary MuSCs to make sure that C2C12 cells are an appropriate model to study STAT3-dependent inhibition of autophagy via PKR and EIF2A.

RE: Albeit MuSCs exhibit a reduced cytoplasm, we managed to measure the STAT3/PKR interaction in the cytoplasmic region, and its disruption upon STAT3i, by PLA validating the use of C2C12 to study STAT3-dependent inhibition of autophagy via PKR and EIF2A (New Supplementary Figure S5B).

5. To characterize transcriptional effects of STAT3 inhibition, the authors performed some RT-PCR assays, which are not very informative. Why not a RNAseq of C2C12 AND primary MuSCs treated with STAT3i? Similar transcriptional effects of STAT3i in C2C12 and primary MuSCs would also justify the use of C2C12 cells as an adequate model for the mechanistic STAT3i experiments.

RE: As suggested by this reviewer, we expanded the analysis of the autophagic related genes in C2C12 and in MuSCs upon STAT3i administration. In the revised version of the MS, we now show the expression of pro-autophagic genes (e.g. *LC3*, *Atg9A*, *Ulk1*, *Bnip3* and *Becn1*) (New Figure 6B) that are shared among C2C12 and MuSCs (New Supplementary Figure S5C).

6. Wildtype control mice are missing in the experiments depicted in Fig. 4E. How can the authors claim 'that STAT3 deletion is sufficient to activate the autophagic flux in MuSCs irrespective of whether they were isolated from young or old mice' without comparing to MuSCs that express Stat3?

RE: In the experiment shown in Figure 4E (now Figure 4G), we employed the Pax7-CreER;Stat3^{fllox/fllox} mice. Upon Tamoxifen administration, Stat3 is ablated specifically in satellite cells and the relative control consists of mice not treated with TMX, therefore with Stat3. Stat3 ablation was controlled by monitoring *Stat3* transcript and STAT3 target gene (*MyoD*) (Figure 3B).

7. The authors claim improved skeletal muscle regeneration of young and aged mice upon STAT3i treatment. The claim is exclusively based on quantification of cross-sectional areas at a single time point after muscle damage, which is not acceptable. The authors have to assess different time points after damage and also stain for embryonic myosin heavy chain as an early marker for regenerating fibers. H&E staining should be supplied as well. Actually, it was not clear from the description when the analysis was done. The authors wrote that muscles were analyzed 5 days after injection of STAT3i. Was the muscle damage done before injecting STAT3i? The extent of muscle regeneration suggests that muscle injury was induced about 10 days before analysis.

RE: Following the reviewer's suggestion, we now show muscle sections stained for H&E (New Supplementary Figure S2C). We included also immunostaining for embryonic MyHC to measure muscle regeneration (New Supplementary Figure S2D), indicating an improved morphology upon STAT3i treatment. We did not analyze the outcome of STAT3i treatment at later time point since we already demonstrated that genetic deletion there is a decrease, while with the pharmacological inhibition an increase in myofiber CSA, as shown in Tierney et al, 2014. In the revised version of the MS, we included a schematic representation of the treatment (New Supplementary Figure S1A and New Supplementary Figure S2A), which shows that the muscle injury was induced 5 days before the analysis.

8. The co-IP shown in Fig. 6B lacks proper quantification, which needs to be added. It would also be helpful to immunoprecipitate PKR and probe with STAT3 antibodies to validate the effects of STAT3i on the interaction between PKR and STAT3. Molecular weight markers are missing for Fig. 6B. What do the different bands on the PKR blot represent?

RE: We now included Co-IP quantification and molecular weight.

Sincerely,
Lucia Latella

May 23, 2024

RE: Life Science Alliance Manuscript #LSA-2023-02503-TR

Dr. Lucia Latella
National Research Council
Institute of Translational Pharmacology
Via del Fosso di Fiorano 64
Via Fosso del Cavaliere 100
Rome, Italy 00143
Italy

Dear Dr. Latella,

Thank you for submitting your revised manuscript entitled "STAT3 inhibition recovers regeneration of aged muscles by restoring autophagy in muscle stem cells". We would be happy to publish your paper in Life Science Alliance pending final revisions necessary to meet our formatting guidelines.

- please be sure that the authorship listing and order is correct
- please add the Twitter handle of your host institute/organization as well as your own or/and one of the authors in our system
- please consult our manuscript preparation guidelines <https://www.life-science-alliance.org/manuscript-prep> and make sure your manuscript sections are in the correct order
- the contributions selected for Pier Lorenzo Puri do not qualify them for authorship. Please either update the contributions in our system and the Author Contributions section of the manuscript or let us know if the author needs to be removed (and added eventually to the Acknowledgments section).
- please add your main and supplementary figure legends to the main manuscript text after the references section
- in the Materials and Methods, please mention approval for the mouse work, and who granted the approval

FIGURE CHECKS:

- please add sizes next to all blots

A. FINAL FILES:

B. MANUSCRIPT ORGANIZATION AND FORMATTING:

Sincerely,

May 24, 2024

RE: Life Science Alliance Manuscript #LSA-2023-02503-TRR

Dr. Lucia Latella
National Research Council
Institute of Translational Pharmacology
Via del Fosso di Fiorano 64
Via Fosso del Cavaliere 100
Rome, Italy 00143
Italy

Dear Dr. Latella,

Thank you for submitting your Research Article entitled "STAT3 inhibition recovers regeneration of aged muscles by restoring autophagy in muscle stem cells". It is a pleasure to let you know that your manuscript is now accepted for publication in Life Science Alliance. Congratulations on this interesting work.

DISTRIBUTION OF MATERIALS:

Again, congratulations on a very nice paper. I hope you found the review process to be constructive and are pleased with how the manuscript was handled editorially. We look forward to future exciting submissions from your lab.

Sincerely,
